# Multimodal mapping of cell types and projections in the central nucleus of the amygdala

Yuhan Wang[1], Sabine Krabbe[2,3], Mark Eddison[1], Fredrick E Henry[1], Greg Fleishman[1], Andrew L Lemire[1], Lihua Wang[1], Wyatt Korff[1], Paul W Tillberg[1], Andreas Lüthi[2], Scott M Sternson[1,4]*

[1]Janelia Research Campus, Howard Hughes Medical Institute, Ashburn, United States; [2]Friedrich Miescher Institute for Biomedical Research, Basel, Switzerland; [3]German Center for Neurodegenerative Diseases (DZNE), Bonn, Germany; [4]Howard Hughes Medical Institute & Department of Neurosciences, University of California, San Diego, San Diego, United States

*For correspondence:
ssternson@health.ucsd.edu

Competing interest: The authors declare that no competing interests exist.

**Abstract** The central nucleus of the amygdala (CEA) is a brain region that integrates external and internal sensory information and executes innate and adaptive behaviors through distinct output pathways. Despite its complex functions, the diversity of molecularly defined neuronal types in the CEA and their contributions to major axonal projection targets have not been examined systematically. Here, we performed single-cell RNA-sequencing (scRNA-seq) to classify molecularly defined cell types in the CEA and identified marker genes to map the location of these neuronal types using expansion-assisted iterative fluorescence in situ hybridization (EASI-FISH). We developed new methods to integrate EASI-FISH with 5-plex retrograde axonal labeling to determine the spatial, morphological, and connectivity properties of ~30,000 molecularly defined CEA neurons. Our study revealed spatiomolecular organization of the CEA, with medial and lateral CEA associated with distinct molecularly defined cell families. We also found a long-range axon projection network from the CEA, where target regions receive inputs from multiple molecularly defined cell types. Axon collateralization was found primarily among projections to hindbrain targets, which are distinct from forebrain projections. This resource reports marker gene combinations for molecularly defined cell types and axon-projection types, which will be useful for selective interrogation of these neuronal populations to study their contributions to the diverse functions of the CEA.

## Editor's evaluation

This study presents a valuable and comprehensive analysis of the molecular identity of neuronal subtypes of the central amygdala, along with their spatial, morphological, and connectivity properties. The evidence supporting the authors' conclusions is compelling and includes the use of rigorous state-of-the-art methodologies for RNA sequencing and spatial profiling as well as a novel approach for integrating molecular identity and axonal projections. This study will interest neuroscientists studying the function of the central amygdala.

## Introduction

Neuronal heterogeneity has been characterized and classified in brains by gene expression, morphology, and connectivity. The increasing number of molecularly defined neuron types revealed by single-cell RNA sequencing (scRNA-seq) accentuates the need for integrated methods to relate

gene expression, cell morphology, and axon projection patterns. Several imaging-based and sequencing-based methods have been developed to allow for spatial gene expression profiling in both thin (~10 μm) and thick (≥100 μm) tissue (*Chen et al., 2015*; *Codeluppi et al., 2018*; *Moffitt et al., 2018*; *Nicovich et al., 2019*; *Qian et al., 2020*; *Shah et al., 2016*; *Wang et al., 2018*), including expansion-assisted iterative fluorescence in situ hybridization (EASI-FISH) (*Wang et al., 2021*). These methods provide spatially resolved molecular composition of neuronal types, but they can also serve as a bridge to link cell types classified using scRNA-seq with functional attributes, such as neuronal projections (*Zhang et al., 2021*) and activity (*Bugeon et al., 2022*; *Lovett-Barron et al., 2020*; *Xu et al., 2020*). Some methods, such as BARseq2 (*Chen et al., 2019*; *Sun et al., 2021*) and MERFISH (*Zhang et al., 2021*), have been developed for mapping neuronal projections in the cortex, but most of these methods have been performed in thin tissue sections, limiting 3D tissue context.

The central nucleus of the amygdala (CEA) (*Cassell et al., 1999*; *Cassell and Gray, 1989*; *Keifer et al., 2015*; *Moscarello and Penzo, 2022*) integrates external and internal sensory information to control motivated behaviors and learning in appetitive (*Cai et al., 2014*; *Carter et al., 2013*; *Hardaway et al., 2019*; *Robinson et al., 2014*) and aversive domains (*Ciocchi et al., 2010*; *Ehrlich et al., 2009*; *Haubensak et al., 2010*; *Wilson et al., 2019*). The CEA also controls a variety of innate responses such as autonomic functions (*Iwata et al., 1987*; *Kapp et al., 1979*), taste valence (*Sadacca et al., 2012*), food and water intake (*Douglass et al., 2017*; *Kim et al., 2017*), jaw movements (*Han et al., 2017*), as well as predatory behavior (*Han et al., 2017*). The CEA is also involved in responses to general anesthetics (*Hua et al., 2020*), addiction (*Domi et al., 2021*; *Torruella-Suárez et al., 2020*; *Venniro et al., 2020*), pain (*Allen et al., 2021*; *Allen et al., 2023*; *Han et al., 2005*; *Okutsu et al., 2017*; *Wilson et al., 2019*), and itch (*Samineni et al., 2021*).

CEA neurons elicit these distinct behavioral functions based on their axon projection patterns. For example, the CEA influences multiple processes associated with threat responses, such as freezing and cardiovascular adaptations, via projections to the ventrolateral periaqueductal gray (vlPAG) (*Tovote et al., 2016*) and the nucleus of the solitary tract (NTS) (*Saha, 2005*). CEA→vPAG projections are also involved in chloroquine-induced pruritic behaviors (*Samineni et al., 2021*). Neurons in the anterior CEA are implicated in predatory hunting responses in rodents via projections to the parvocellular reticular formation (PCRt) and the vlPAG (*Han et al., 2017*). CEA projections to other hindbrain regions (*Veening et al., 1984*), such as the parabrachial nucleus (PBN), modulate food intake (*Douglass et al., 2017*), alcohol consumption (*Bloodgood et al., 2021*; *Torruella-Suárez et al., 2020*), and pain responses and aversion (*Allen et al., 2021*; *Allen et al., 2023*; *Han et al., 2005*; *Han et al., 2015*; *Okutsu et al., 2017*; *Wilson et al., 2019*). In addition, the CEA→lateral SN (lateral substantia nigra) pathway promotes learned behavioral responses to salient stimuli (*Lee et al., 2005*; *Steinberg et al., 2020*).

The CEA consists of primarily GABAergic neurons that are organized within at least three subdivisions. In the classical view (*Duvarci et al., 2011*), the lateral central amygdala (CeL) is a primary target for external and internal sensory inputs that are processed and passed to the medial central amygdala (CeM), a major output nucleus projecting to hindbrain autonomic and motor control areas. For instance, activation of CeM elicits freezing behavior (*Ciocchi et al., 2010*), which is gated by disinhibition of a local inhibitory projection from CeL to CeM (*Haubensak et al., 2010*). However, long-range projections from the CeL have been reported (*Cassell et al., 1999*; *Herry and Johansen, 2014*; *Li and Sheets, 2018*; *Veening et al., 1984*; *Ye and Veinante, 2019*), indicating direct influence of this subregion on downstream areas. In addition, the capsular central nucleus (CeC), a subdivision at the far lateral edge of the middle and posterior CEA and the anterior lateral portion of CEA, receives distinct axonal inputs (*Cassell et al., 1999*) and contributes to controlling defensive behaviors (*Kim et al., 2017*) and appetite (*Carter et al., 2013*). In addition to distinct projection patterns, neurons in different CEA subnuclei also showed distinct electrophysiological properties and morphology (*Li et al., 2022*; *Li and Sheets, 2018*; *Mork et al., 2022*), with additional functional heterogeneity observed along the rostrocaudal topographic axis (*Bowen et al., 2022*).

The CEA also contains multiple molecularly defined neuron populations that have been investigated for their role in appetitive and defensive behaviors. Previously used marker genes for CEA cell types include protein kinase C delta (*Prkcd*), somatostatin (*Sst*), corticotropin-releasing hormone (*Crh*), tachykinin-2 (*Tac2*), or neurotensin (*Nts*). *Prkcd* is expressed in a set of CEA neurons that reduce freezing, promote extinction learning, suppress appetite when activated (*Cai et al., 2014*; *Haubensak*

*et al., 2010*; *Kim et al., 2017*), and elicit defensive behaviors when inhibited (*Ciocchi et al., 2010*). CEA *Prkcd* neurons are also associated with drug craving (*Venniro et al., 2020*) and alcohol addiction (*Domi et al., 2021*). *Sst*, *Crh*, and *Tac2* neurons have been associated with the acquisition and expression of conditioned fear responses such as freezing or flight (*Andero et al., 2016*; *Andero et al., 2014*; *Gafford and Ressler, 2015*; *Li et al., 2013*; *Sanford et al., 2017*; *Yu et al., 2016*). In addition, *Nts*-expressing neurons are involved in feeding and reward-related behaviors (*Kim et al., 2017*; *Torruella-Suárez et al., 2020*). Additional work has demonstrated that some of these markers (*Sst, Crh, Nts*) have a considerable degree of overlap in the same neurons, whereas others are distinct (e.g., *Prkcd*) (*McCullough et al., 2018b*). However, the molecular diversity of CEA neurons has been incompletely examined. In addition, it has been difficult to establish the correspondence of CEA axonal projections with its many molecularly defined cell types. Therefore, the functional significance of CEA subpopulations defined by both anatomical and molecular characteristics remains largely uncharacterized.

To increase understanding of the organization of the CEA, it is important to systematically classify the molecularly defined neuron types in the CEA as well as their anatomical locations in CEA subnuclei and major projection pathways. Here, we performed single-cell RNA-sequencing (scRNA-seq) on neurons from the CEA and used these data to define molecularly defined neuron types. We then integrated retrograde axonal tracing with EASI-FISH (*Wang et al., 2021*) in 100-µm-thick tissue sections to determine the molecular identity and spatial distribution of neurons that project to several important CEA output targets. This produced a molecular parcellation of the CEA, comprising multiple cell populations with complex projection patterns.

## Results

### Molecularly defined cell types in the CEA based on scRNA-seq

We used scRNA-seq to profile gene expression diversity in CEA neurons (*Figure 1—figure supplement 1A–D*) and identified 13 transcriptomic neuronal types from the CEA (*Figure 1A and D*, also see 'Materials and methods' and online portal). Consistent with previous descriptions, neurons in this region expressed inhibitory markers, such as glutamate decarboxylase 1 and 2 (*Gad1* and *Gad2*), vesicular GABA transporter (*Slc32a1* encoding Vgat), as well as many neuropeptides and neuromodulatory receptors (*Figure 1B*, *Figure 1—figure supplement 2*).

Clustering analysis revealed two CEA neuronal classes (class 1 and class 2), primarily distinguished by the expression of *Ppp1r1b*, encoding DARPP-32, a target of dopamine and glutamate signaling (*Fernandez et al., 2006*; *Figure 1C*). Class 1 contained a family of *Sst/Pdyn* co-expressing neurons that was comprised of two types distinguished by additional co-expression of *Crh*, *Tac2*, *Nts*, and *Vipr2* in seq-c7 and lacking these genes in seq-c10. Another population, seq-c5, expressed mRNA encoding D2-receptor (*Drd2*) and voltage-gated sodium channel β subunit, *Scn4b*. Seq-c8 was the primary *Prkcd*-expressing (97%, 107.0 ± 6.9 UMIs/cell) population that co-expressed *Cartpt* at high levels (87.1%, 158.0 ± 16.8 UMIs/cell). This cluster was closely related to seq-c6, in which some neurons also expressed *Prkcd* (48%, 69.9 ± 10.0 UMIs/cell) and only 20% co-expressed *Cartpt* at lower levels (65.7 ± 16.6 UMIs/cell), but seq-c6 differed from seq-c8 by expression of other genes, such as *Cyp26b1*, *Crym*, and *Penk* (*Figure 1—figure supplement 2*). Although calcitonin receptor-like (*Calcrl*) has been proposed as a distinguishing marker gene between *Prkcd* neuron types (*Kim et al., 2017*), it was expressed in a subset of cells in both seq-c6 (9.1% of neurons) and seq-c8 (36% of neurons). Seq-c1 was a *Prkcd*-negative population that also contained fewer *Ppp1r1b*-expressing neurons (30%) (*Figure 1—figure supplement 2*); instead, it was characterized by higher levels of histone variant, *H2afz* and thioredoxin 1 (*Txn1*).

Class 2 neurons lacked *Ppp1r1b* and many expressed *Nefm*. Class 2 contained three well-separated cell types: seq-c9 expressed vitamin D receptor (*Vdr*) but had low *Nefm*, seq-c3 co-expressed *Gpx3* and *Gal*, and seq-c11 co-expressed *Sema3c*, *Tac1*, *Sst*, and *Dlk1* (*Figure 1A and C*). The remaining four cell types in class 2 showed greater similarity (*Figure 1C*). Seq-c2 and seq-c4 expressed *Ppia*, *Actg1*, and *Aldoa*, while seq-c12 and seq-c13 did not. Seq-c2 and seq-c4 differed in their expression of *Tac2*. Seq-c13 expressed *Dlk1* and *Cyp26b1*, which were absent from seq-c12. In addition, *Htr2a* has been previously used as a marker gene in the CEA. In this scRNA-seq dataset, very few cells with *Htr2a* RNA expression were detected (*Figure 1—figure supplement 2*), consistent with some

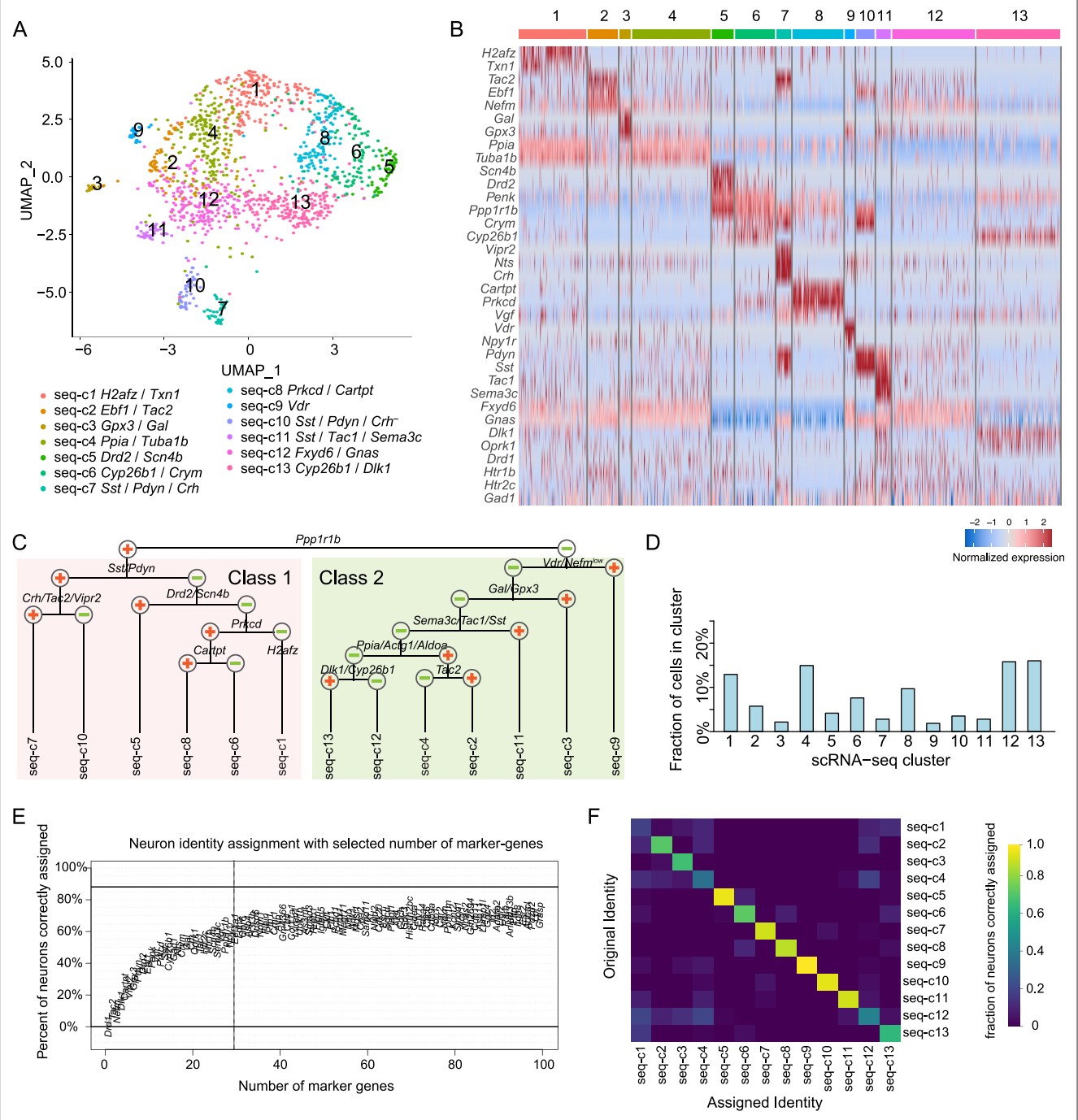

**Figure 1.** Central nucleus of the amygdala (CEA) single-cell RNA-sequencing (scRNA-seq) data analysis. (**A**) UMAP for molecularly defined neuron clusters in the CEA, with cell types color-coded by scRNA-seq clusters. A total of 1,643 cells were subjected to single-cell RNA sequencing, with an average depth of 6,611,566 ± 92,440 (mean ± SD) reads per cell. Among them, 1,393 CEA neurons were identified. (**B**) Heatmap of marker genes from scRNA-seq clusters. Colormap indicates z-score normalized expression. (**C**) Dendrogram representing transcriptional relationships of molecularly defined neuronal types from scRNA-seq. (**D**) Fraction of neurons in each scRNA-seq cluster. (**E**) Percent of correctly mapped neurons with increasing numbers of marker genes. The dotted vertical line indicates the cutoff for marker genes selected in this study. The top 100 most differentially expressed marker genes were included for this analysis. (**F**) Heatmap showing the fraction of neurons that were correctly assigned to their original scRNA-seq cluster using 29 selected marker genes.

The online version of this article includes the following figure supplement(s) for figure 1:

**Figure supplement 1.** Microdissection of central nucleus of the amygdala (CEA) tissue for single-cell RNA sequencing (scRNA-seq).

**Figure supplement 2.** UMAP showing the expression of selected genes from single-cell RNA sequencing (scRNA-seq) data.

*Figure 1 continued on next page*

*Figure 1 continued*

**Figure supplement 3.** Integration of central nucleus of the amygdala (CEA) single-cell RNA-sequencing (scRNA-seq) data from two datasets.

other reports (*Hardaway et al., 2019*; *Kim et al., 2017*). To determine how our scRNA-seq dataset compared with recently published CEA scRNA-seq data (*O'Leary et al., 2022*), we integrated the two datasets via canonical correlation analysis (CCA) (*Figure 1—figure supplement 3*) and found that all neurons from the O'Leary et al. dataset mapped with neuronal clusters from our dataset, indicating that our dataset covered the molecular diversity in the CEA. Analysis of the newly combined data also did not affect clustering of most neurons from our original scRNA-seq dataset (*Figure 1—figure supplement 3D*).

For seq-c1, c4 and c12, although differentially expressed genes were identified, these genes had low selectivity in the CEA based on scRNA-seq analysis (the fraction of neurons with marker gene expression in the cluster versus expression of that gene in the rest of the populations, pct.1 and pct.2 in *Supplementary file 1*). Broad expression of these genes was also apparent in Allen Brain Atlas ISH images (*Lein et al., 2007*; *Figure 1—figure supplement 1F–H*). Because of this, we de-prioritized these clusters when selecting marker genes. We selected a set of 23 marker genes whose combinations were used to define major CEA molecular types. Additionally, we included *Gad1* to distinguish CEA neurons from non-neurons and five additional differentially expressed neuromodulatory GPCRs (*Npy1r*, *Drd1*, *Drd2*, *Htr1b*, and *Htr2c*) that showed selective expression patterns in the CEA. Based on selected marker genes (29 total, *Supplementary file 2*) expression, more than 60% of neurons correctly mapped to their original molecular identity, and further increasing the number of marker genes did not substantially improve this (*Figure 1E*). Also, 10 out of 13 scRNA-seq clusters could be identified by these marker genes, as defined by greater than 50% neurons mapped correctly from these clusters (*Figure 1F*). The unmapped scRNA-seq clusters lacked highly specific marker genes (seq-c1, seq-c4, seq-c12). Instead, seq-c4 and seq-c12 can be considered as *Nefm*-expressing CEA neurons that lack additional marker genes (*Nefm* could be substituted with *Fxyd6*), which is also consistent with our analysis of the combined scRNA-seq data (*Figure 1—figure supplement 3E*).

## EASI-FISH with retrograde tracing

To map the spatial distribution and axon projection patterns of molecularly defined neurons identified by scRNA-seq, we developed methods to combine EASI-FISH using 29 marker genes with 5-plex retrograde tracing using fluorophore-conjugated cholera toxin subunit B (CTb) and hydroxystilbamidine (FluoroGold) (*Saleeba et al., 2019*), which showed good labeling in the CEA (*Figure 2—figure supplement 1*). We selected five brain regions that have been previously identified as CEA targets/effectors and are involved in appetitive and defensive behaviors: the bed nucleus of the stria terminalis (BNST), lateral substantia nigra (lateral SN), ventrolateral PAG (vlPAG), parabrachial nucleus (PBN), and parvocellular reticular formation (PCRt) (*Cai et al., 2014*; *Douglass et al., 2017*; *Fadok et al., 2017*; *Han et al., 2017*; *LeDoux et al., 1988*; *Steinberg et al., 2020*; *Tovote et al., 2016*). In each mouse, retrograde tracers with distinct fluorophores were injected into the five sites to label CEA neurons that project to these regions (*Figure 2—figure supplement 2A and B*; also see 'Materials and methods'). We aimed to maximize the number of retrogradely labeled CEA projecting neurons to each brain area with injections that encompassed most of each brain region but limited spread outside of the targeted areas. However, injections into the PBN included some adjacent medial structures, such as the locus-coeruleus and peri-locus coeruleus. vlPAG injections also labeled varying portions of dorsal PAG along the injection pipette track (*Figure 2—figure supplement 2A and B*).

We developed a two-stage protocol to combine CTb and fluorogold labeling with EASI-FISH. First, we imaged retrogradely labeled neurons and DAPI-stained nuclei from CEA-containing brain sections (100 μm) that were cleared using 8% sodium dodecyl sulfate (SDS) (*Nicovich et al., 2019*) and refractive index matched to allow for imaging of fluorophores throughout the 100-μm-thick tissue using confocal microscopy (see 'Materials and methods,' *Figure 2—figure supplement 3A*). Next, we processed the same tissue sections for EASI-FISH, which is a method based on expansion microscopy that facilitates high-quality imaging of mRNA in thick tissue sections (*Wang et al., 2021*). Protease treatment in the EASI-FISH procedure, which is necessary for tissue clearing and tissue expansion, also removes CTb proteins and Fluorogold (*Figure 2—figure supplement 3B*), thus avoiding fluorophore

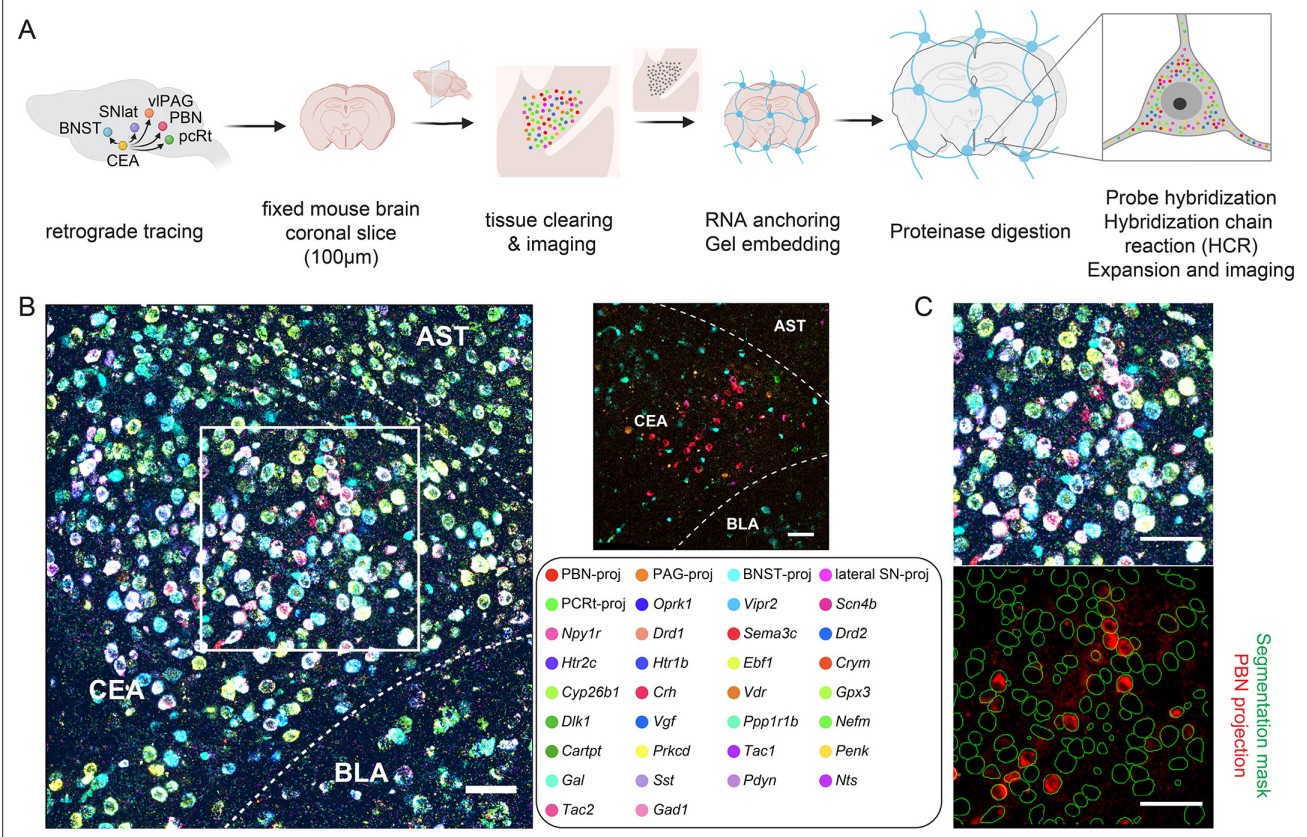

**Figure 2.** Method to combine expansion-assisted iterative fluorescence in situ hybridization (EASI-FISH) with projection class mapping in the central amygdala. (**A**) Schematics of the procedure. Fluorophore-labeled cholera toxin subunit B (CTbs) and fluorogold were used to retrogradely trace neurons in the central nucleus of the amygdala (CEA) that project to bed nucleus of the stria terminalis (BNST), lateral substantia nigra (SN), ventrolateral periaqueductal gray (vlPAG), parabrachial nucleus (PBN), and parvocellular reticular formation (PCRt). Confocal images were collected to identify projection classes in the CEA. Retrograde tracer fluorescence signals were then eliminated from tissue sections and the EASI-FISH procedure was performed as described in *Wang et al., 2021*. (**B**) Representative image showing the molecularly defined and projection-defined cell types in the CEA (left). Right: same as left, with projection types only. Dotted line: borders between the CEA and surrounding brain areas. AST: amygdalostriatal transition area; BLA: basolateral amygdala. Scale bars: 50 µm. (**C**) Zoom-in of the box in (**B**) showing registration between confocal image and EASI-FISH images (top). This allows the extraction of projection signals using EASI-FISH segmentation mask (bottom). Scale bars: 50 µm.

The online version of this article includes the following figure supplement(s) for figure 2:

**Figure supplement 1.** Retrograde tracer labeling in the central nucleus of the amygdala (CEA).

**Figure supplement 2.** Retrograde tracer labeling of central nucleus of the amygdala (CEA) neurons for five projection targets.

**Figure supplement 3.** Method validation.

ambiguity between retrograde labeling and FISH (*Figure 2A*). The tissue-clearing procedure did not compromise RNA quality (*Figure 2—figure supplement 3C*).

EASI-FISH measures gene expression using multiple rounds of FISH. To align image volumes across rounds, EASI-FISH relies on DAPI-stained RNA that conveniently labels the cytoplasm (cytoDAPI) after DNase treatment, which is used to strip oligonucleotide probes between FISH rounds but also eliminates DAPI staining of nuclei. For rapid volumetric imaging while limiting photobleaching, EASI-FISH-processed tissue is imaged using selective plane illumination microscopy (SPIM or 'light-sheet' microscopy). To register confocal image volumes that contained information about neuronal projections with SPIM image volumes that contained information about gene expression, we developed an experimental and computational modification to the EASI-FISH procedure (see 'Materials and methods'). For this, an additional staining and imaging round (called round 0 because marker genes were not probed) was included to 'link' the confocal and the EASI-FISH image volumes at the beginning of the EASI-FISH experiment, where nuclei were labeled with DAPI, and the surrounding cytosolic region of a cell was marked with ribosomal RNA (rRNA) probes (*Figure 2—figure supplement*

*3D*). The nuclear DAPI images were used to register the first EASI-FISH image volumes with confocal images, while the cytosolic rRNA-stain was used to register with cytoDAPI in subsequent EASI-FISH rounds that probed for CEA marker genes (*Figure 2—figure supplement 3E*). Using this procedure, we acquired axon projection information targeting five downstream brain regions along with expression levels of 29 marker genes in individual neurons from the same tissue (*Figure 2B*). 29-plex marker gene expression was measured and computationally extracted using the previously described EASI-FISH procedure and data processing pipeline (*Wang et al., 2021*) (https://github.com/JaneliaSciComp/multifish). This includes automatically generated three-dimensional segmentation masks for every cell in the tissue volume. High-quality confocal-to-SPIM image registration using this pipeline permitted us to apply the high-resolution EASI-FISH segmentation masks to confocal images to extract fluorescence intensities from CTb or FluoroGold that indicated the neuronal projection types (*Figure 2C*). Importantly, this is simpler and more accurate than generating and reconciling separate segmentation masks for the same specimen across image volumes that were acquired using different modalities (confocal and SPIM).

Low expressor marker genes (*Oprk1*, *Scn4b,* and *Vipr2*) were reprobed at the last round of EASI-FISH to evaluate RNA loss during the procedure. We found that greater than 70% of RNAs were retained with EASI-FISH after 2 months (*Figure 2—figure supplement 3F and G*). Of note, we were still able to detect nearly 60% of RNA in samples after 1 year (*Figure 2—figure supplement 3F and H*).

## Identification of EASI-FISH molecular clusters

We analyzed the spatial organization of molecularly defined and projection-defined neurons in the CEA and surrounding regions at multiple levels (anterior, A; middle, M; and posterior, P) from three animals. Tissue from two animals was used for the initial analysis, and data from the third animal (ANM #3) was used for cross-validation.

After EASI-FISH image processing, RNA-spot counting, and cell boundary extraction, cells were clustered based on marker gene expression (*Figure 3—figure supplement 1A*) and non-neurons were removed based on small cell size and lack of CEA neuronal marker gene expression, such as *Gad1* (*Figure 3—figure supplement 1B*; also see 'Materials and methods'). We obtained 33,139 neurons from six samples (ANM#1A: 5,604 neurons; ANM#1M: 4,286 neurons; ANM#1P: 4,728 neurons; ANM#2A: 5,684 neurons; ANM#2M: 7,342 neurons; ANM#2P: 5,495 neurons) (*Figure 3—figure supplement 1C* and *Supplementary file 3*). We used a de novo approach to identify cell types in the CEA and then correlated identified EASI-FISH clusters with scRNA-seq clusters based on 29-plex marker gene expression (online portal). Most neurons (91.4%) could be classified into 21 distinct molecular clusters (MCs) based on clustering analysis, with the remaining *Gad1*-expressing neurons (8.6%, denoted as MC-22) showing low expression of the selected marker genes (*Figure 3A and B*). Among the 21 molecular clusters, 17 were from the CEA, 1 cluster was from the basolateral amygdala (BLA) expressing *Oprk1* (MC-8), 1 cluster was the intercalated neurons (ITCs) expressing *Drd1/Penk* (MC-21), and 2 clusters were from amygdalostriatal transition area (AST) expressing *Drd1* (MC-7) or *Drd2* (MC-20) (*Figure 3—figure supplement 2*).

Clusters identified by EASI-FISH showed good correspondence with scRNA-seq (*Figure 3C*, *Figure 3—figure supplement 1E*, *Figure 3—figure supplement 1—source data 1*). Neurons from the two top-level classes defined by scRNA-seq were spatially separate (*Figure 3D*), with MCs from class 1 enriched in the lateral and capsular parts of the CEA and most neurons from class 2 enriched in the medial part of the CEA (*Figure 3E*). We mapped all 17 EASI-FISH clusters in the CEA to the 13 scRNA-seq clusters. For example, consistent with scRNA-seq data, we identified two major *Sst/Pdyn* co-expressing populations using EASI-FISH. One (MC-2) co-expressed *Vipr2*, *Nts*, *Crh*, and *Tac2*, corresponding to seq-c7, while MC-3 lacked expression of these genes, corresponding to seq-c10. Because many neurons were profiled with EASI-FISH, this allowed us to uncover intra-cluster heterogeneities, which lead to further subdivision of scRNA-seq clusters. For example, seq-c3 was subdivided into *Gal*+ (MC-9) and *Gal*- (MC-19) populations, and seq-c11 was subdivided into *Tac1*+ (MC-11) and *Tac1*- (MC-18) populations. In addition, seq-c5, the major *Drd2*-expressing population mapped to two EASI-FISH clusters in the CEA, one spanning the CeC and AST (MC-5) and one enriched in CeM (MC-14).

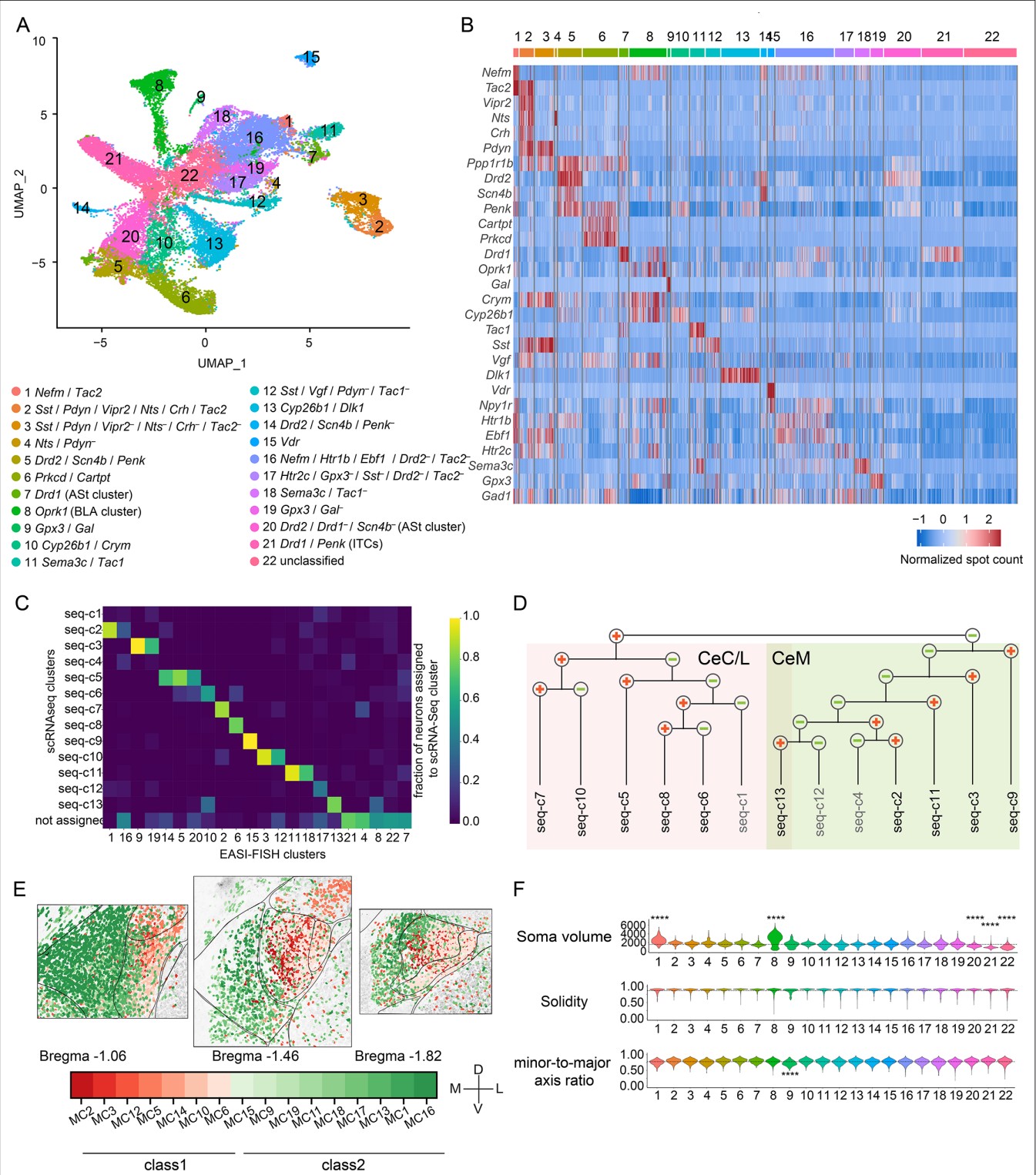

**Figure 3.** Central nucleus of the amygdala (CEA) expansion-assisted iterative fluorescence in situ hybridization (EASI-FISH) gene expression profiling. (**A**) UMAP for molecularly defined EASI-FISH clusters in the CEA. (**B**) Heatmap of 29 FISH marker genes in EASI-FISH clusters. Colormap indicates z-score normalized spot count. (**C**) The proportion of neurons from EASI-FISH clusters assigned to scRNA-seq clusters based on cross-correlation of marker gene expression. (**D, E**) Based on FISH cluster assignment to scRNA-seq clusters and their spatial location, scRNA-seq clusters belonging to separate branches of the dendrogram (from *Figure 1C*) mapped to separate CEA subregions. (**D**) Dendrogram showing the gene-expression relationships of scRNA-seq clusters, same as *Figure 1C*, with mapped subregions colored (CeC and CeL: red, CeM: green). (**E**) Spatial distribution

*Figure 3 continued on next page*

*Figure 3 continued*

of EASI-FISH clusters mapped to scRNA-seq clusters on the separate branches of the dendrogram (class 1: red; class 2: green). (**F**) Morphological properties of somata in EASI-FISH clusters. Top: soma volume; middle: solidity; bottom: the ratio between the minor axis and the major axis of an ellipse fitting the cell outline. Dotted lines: population average. p-values with medium and high effect sizes (Cohen's d>0.5, or rg>0.28) are shown. *p<0.05, **p<0.01, ***p<0.001, ****p<0.0001.

The online version of this article includes the following source data and figure supplement(s) for figure 3:

**Figure supplement 1.** Central nucleus of the amygdala (CEA) expansion-assisted iterative fluorescence in situ hybridization (EASI-FISH) data analysis.

**Figure supplement 1—source data 1.** Table summarizing the correspondence between expansion-assisted iterative fluorescence in situ hybridization (EASI-FISH) clusters and single-cell RNA-sequencing (scRNA-seq) clusters, related to *Figure 3—figure supplement 1E*.

**Figure supplement 2.** UMAP showing marker gene expression as measured by expansion-assisted iterative fluorescence in situ hybridization (EASI-FISH).

## Somatic size and shape of CEA cell types

Because the EASI-FISH analysis pipeline provides high-quality three-dimensional neuronal soma segmentation, we also evaluated the somatic shape of the molecularly defined CEA cell types. Most neurons in the CEA share a similar soma size and shape (*Figure 3F*), consistent with the previous report from rat (*Cassell and Gray, 1989*). Only CEA neurons in MC-1 (*Nefm/Tac2*) from CeM were significantly larger (mean somatic volume: 2506.9 ± 818.5 µm$^3$) than the mean CEA neuron soma size (1581.3 ± 744.6 µm$^3$) (p<0.0001). These neurons were also larger than the *Tac2*-expressing population from CeL (MC-2: 1829.4 ± 413.6 µm$^3$, p<0.0001). MC-1 neurons with larger somata expressed membrane proteins, such as *Cadps2*, *Gpr101*, and *Tacr3*, according to scRNA-seq data (*Figure 3—figure supplement 1F*). Another population, MC-9 (*Gpx3/Gal*) from CeM, had a significantly lower minor-to-major axis ratio, indicating that they are more elongated than most CEA cell populations. These MC-9 neurons overexpressed distinct membrane proteins, such as *Syt17*, *Cdh11*, and *Grin3a* (*Figure 3—figure supplement 1G*). CEA neurons that only expressed *Gad1* but lacked specific marker genes (MC-22) also had significantly smaller somata than the mean of CEA neurons.

Neuronal soma size heterogeneity was also observed in surrounding areas. The BLA (MC-8) has many large neurons and some small neurons (mean somatic volume: 2736.0 ± 1291.5 µm$^3$), which is consistent with the previous report of principal neurons with large somata, and interneurons with smaller somata (*McDonald, 2020*; *Figure 3F*). The ITCs (MC-21) were the smallest (906.5 ± 317.3 µm$^3$) and the D2-receptor-expressing neurons in the AST (MC-20) were also smaller (1229.3 ± 372.6 µm$^3$) than most CEA neurons (p<0.0001). Because our segmentation approach is largely restricted to neuronal somata, future studies will be needed to determine how this heterogeneity in soma size and shape relates to complete neuronal morphology and functions.

## Parcellation of CEA

Canonical subdivisions of the CEA in mouse are derived from neuroanatomical studies in rats, where three sub-nuclei (CeM, CeL, and CeC) have been identified based on cytoarchitecture, connectivity, and selected neuropeptide expression, such as enkephalin (*Cassell et al., 1999*). In mice, these boundaries can be difficult to discern reliably and precisely (*Ye and Veinante, 2019*). A consistent parcellation of the CEA would be facilitated by molecular markers that are associated with different subregions. These genes need not be expressed in every neuron within a particular region but, instead, have a combinatorial expression pattern from which the boundaries are apparent. To identify these marker gene combinations, we examined the spatial variation of marker gene expression in the CEA and its surrounding region. Principal component analysis (PCA) of the expression patterns of 29 marker genes identified spatial patterns that explained the greatest proportion of molecular variance in the region. As expected, this revealed regional boundaries with surrounding brain areas (CEA, BLA, PAL, AST, and ITCs) (*Figure 4—figure supplement 1A and B*), consistent with previous findings (*Zirlinger et al., 2001*). Within the CEA, most PCs associated with the largest gene expression variation corresponded to differences between capsular/lateral zones and the medial zone (CeC/CeL and CeM), consistent with our scRNA-seq hierarchical analysis (*Figure 3D*). PCA revealed *Ppp1r1b* and *Nefm* as marker genes with large loadings in the top 4 PCs that demarcate these subregions along the anterior–posterior extent of the CEA, with *Ppp1r1b* expressed laterally and *Nefm* medially (*Figure 4—figure supplement 1A and B*). Additional spatial variation in gene expression was observed in the lateral

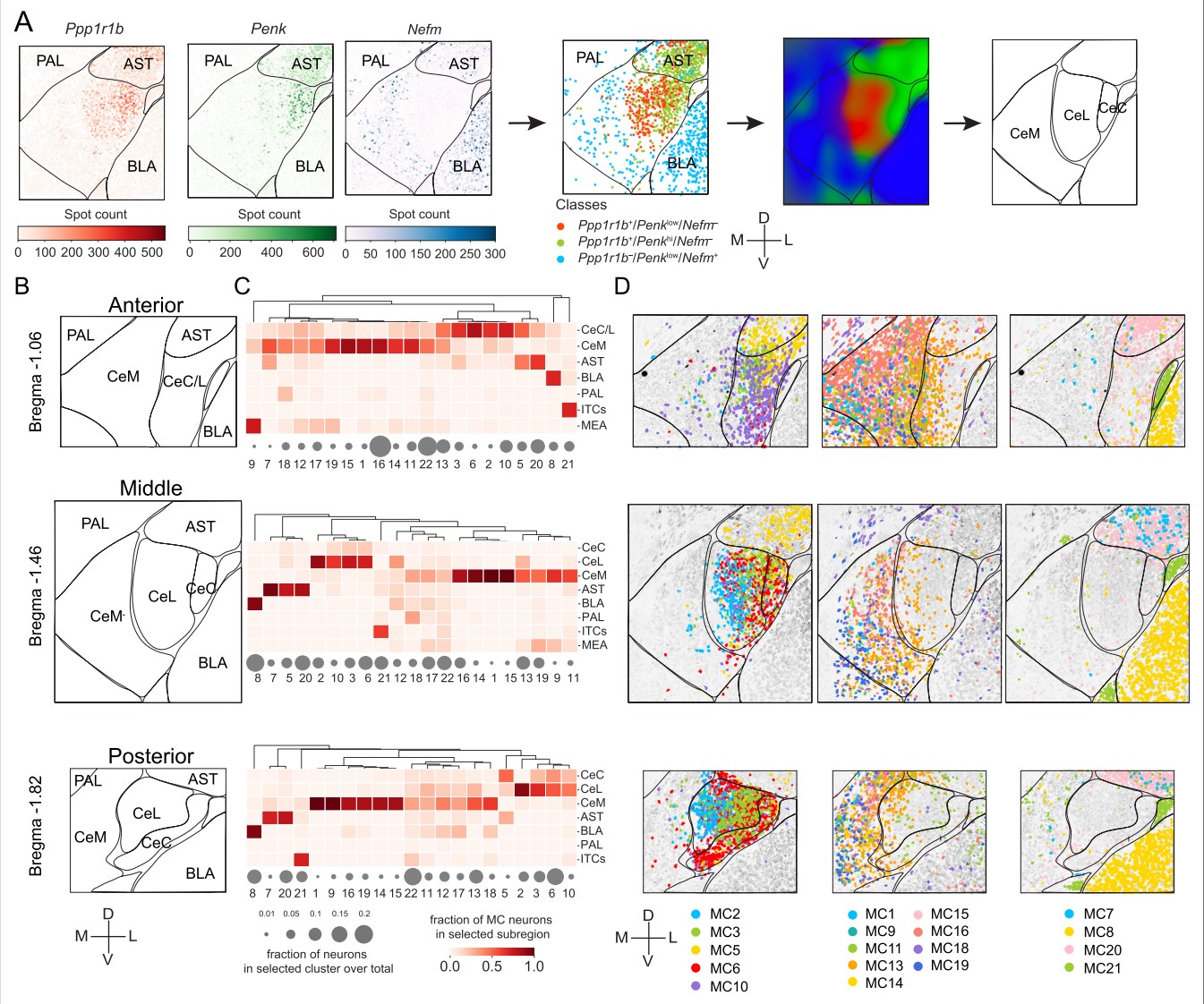

**Figure 4.** Expansion-assisted iterative fluorescence in situ hybridization (EASI-FISH) spatial analysis. (**A**) Representative diagrams detailing the anatomical parcellation procedure in the central nucleus of the amygdala (CEA). First, marker genes (*Ppp1r1b*, *Penk*, and *Nefm*) identified from principal component analysis (PCA) were used to classify neurons and create the anatomical parcellation using the probabilistic Gaussian process classification. BLA: basolateral amygdala; AST: amygdalostriatal transition area; PAL: pallidum; ITCs: the intercalated cells of the amygdala; MEA: medial amygdala. (**B**) Anatomical parcellations from anterior (top), middle (middle), and posterior (bottom) CEA from one animal (ANM #1). (**C**) EASI-FISH cluster enrichment in the parcellated subregions. Gray circles: fractions of neurons profiled that belong to selected clusters. Color bar: fraction of MC neurons in each subregion. (**D**) Spatial distribution of molecularly defined neuron types enriched in parcellated subregions. Colors represent cluster identity. All neurons are colored in light gray in the background. Scale bars in (**A**, **B**, **D**): 200 μm. M: medial, L: lateral, D: dorsal, V: ventral.

The online version of this article includes the following source data and figure supplement(s) for figure 4:

**Source data 1.** Spatial distribution of central nucleus of the amygdala (CEA) clusters.

**Figure supplement 1.** Central nucleus of the amygdala (CEA) spatial analysis.

**Figure supplement 2.** Spatial parcellation in the central nucleus of the amygdala (CEA).

**Figure supplement 3.** Molecularly defined neuron types enriched in central nucleus of the amygdala (CEA) subregions.

**Figure supplement 4.** Expression of selected neuromodulatory GPCRs in the central nucleus of the amygdala (CEA).

part of the CEA, especially in the more posterior sections. Instead of a clear separation, a gradient of gene expression was observed in PC2-4 in most samples, with *Penk* showing large loadings on these PCs (***Figure 4—figure supplement 1A and B***). In light of this, we used *Penk*, which has been used previously to subdivide CeC from CeL (***Cassell et al., 1986***), as an additional marker gene.

Using *Ppp1r1b*, *Nefm,* and *Penk*, we parcellated the CEA into three subdomains using probabilistic Gaussian process classification (*Figure 4A*; see 'Materials and methods'). This offers a systematic approach to define boundaries between CEA subregions in the mouse based on three marker genes. The CeC and CeL separation was not apparent in the anterior sections. Thus, we denoted this anterior region as CeC/L (*Figure 4B*, *Figure 4—figure supplement 2*). Seventeen molecularly defined neuron types were spatially enriched in one or more CEA subregions (*Figure 4C*, *Figure 4—figure supplement 2*, *Figure 4—figure supplement 3*).

## CeM

We found that nine molecularly defined neuron types were enriched in the CeM (*Figure 4D*, *Figure 4—source data 1*). MC-16 (*Nefm/Htr1b/Ebf1/Drd2⁻/Tac2⁻*) was the most abundant cell type in the CeM, with the highest density in the anterior part. MC-16 is a *Nefm*-expressing cluster (79.0%) that can be largely defined by *Htr1b* and *Ebf1*, genes with low expression levels. MC-16 also lacked expression of marker genes that were present in other CeM MCs, such as *Drd2* and *Tac2*. Many neurons in this cluster expressed *Npy1r* (50.3%) and *Drd1* (33.3%). Neurons from MC-16 corresponded to two scRNA-seq clusters, seq-c2 (54%) and seq-c4 (40%), but they have lower cross-correlation coefficient (0.54 ± 0.09) with corresponding scRNA-seq clusters. MC-13, which co-expressed *Dlk1* and *Cyp26b1*, was the most abundant cell type in the posterior part of the CeM and occupied a mostly separate domain lateral to MC-16 in anterior CEA that overlapped the CeM and CeL boundary. This cell type lacked expression of *Ppp1r1b* and corresponded to seq-c13. The two intermediate abundance cell types were neurons expressing *Sema3c* (MC-11 and MC-18). These neurons corresponded to seq-c11 but were split into two FISH clusters intermingled in CeM: MC-11 (*Sema3c/Tac1*), which was the primary *Tac1*-expressing type in the CEA with many neurons also expressing *Dlk1* (74.1%) and *Sst* (51.9%), and MC-18 (*Sema3c/Tac1⁻*) scattered among CeM, CeL, BLA, and PAL. Five low-abundance CeM cell types were found. MC-1 (*Nefm/Tac2*) neurons were the primary CeM *Tac2* cell type, which also had the largest cell bodies in the CEA (*Figure 3F*) and co-expressed a variety of neuromodulatory receptors, such as kappa-opioid receptor (*Oprk1*), neuropeptide Y receptor 1 (*Npy1r*), and serotonin receptor (*Htr1b*). MC-14 (*Scn4b/Penk⁻*) was the primary D2 receptor-expressing (*Drd2*) (90%) cell type in the CeM, and many neurons in this cluster also co-expressed *Htr1b* (79.1%). MC-15 was a vitamin D receptor (*Vdr*) population distributed throughout CeM that co-expressed *Npy1r* (74.1%). Two *Gpx3*-expressing cell types, MC-9 (*Gpx3/Gal*) and MC-19 (*Gpx3/Gal⁻*), which can be distinguished by *Gal* expression, were in the ventral portion of CeM and also in the adjacent medial amygdala (MEA).

## CeC and CeL

The primary CeC and CeL cell type at middle and posterior CEA levels was MC-6 (*Prkcd/Cartpt*). The primary anterior CeC cell type was MC-10 (*Cyp26b1/Crym*), with a small fraction of MC-10 neurons scattered in CeM. MC-6 corresponded to seq-c8, and MC-10 corresponded to seq-c6, which was the main category of *Prkcd*-expressing neurons in the CEA (*Figure 3—figure supplement 1D*). In contrast to many CeC and CeL cell types, MC-10 had low *Ppp1r1b* expression. Two *Sst/Pdyn* cell types (MC-2 and MC-3), corresponding to scRNA-seq clusters, seq-c7 and seq-c10, respectively, were also spatially separated, with MC-2 more medial and uniquely marked by *Vipr2*. MC-2 (*Vipr2*) was the most medial CeL cell type and co-expressed a variety of CeL neuropeptides, including *Pdyn*, *Tac2*, *Nts*, *Crh*, and *Sst* in various proportions (*Supplementary file 4*), and was primarily restricted to middle and posterior levels of CeL (*Figure 4—source data 1*). MC-3 (*Sst/Pdyn/Vipr2⁻*) was found across all levels of CeL, including medial CeC in anterior sections. MC-5 (*Drd2/Scn4b/Penk*) spanned the AST and the CeC. It was spatially enriched in the anterior part of this region and was mostly absent in the CeL.

Although most cell types were localized to CeM, CeC, or CeL, we found that MC-12 (*Sst/Vgf/Pdyn⁻/Tac1⁻*) and MC-17 (*Htr2c/Gpx3⁻/Sst⁻/Drd2⁻/Tac2⁻*) were broadly distributed across the CEA subnuclei. MC-4 (*Nts/Pdyn⁻*) was a low-abundance *Nts* cell type that was scattered in the ventral part of the CeM. We also examined the distribution of neuromodulatory GPCRs (*Figure 4—figure supplement 4A and B*). The serotonin receptors *Htr1b* and *Htr2c* were broadly expressed in many molecularly defined cell types, with the highest average *Htr1b* and *Htr2c* expression in MC-1 and MC-17, respectively. *Htr1b* was also expressed in a few other clusters, such as MC-5, MC-14, and MC-16. Neuropeptide Y receptor 1 (*Npy1r*) was expressed in a restricted group of CeM clusters (MC-1, MC-15, and MC-16). The dopamine receptors, *Drd1* and *Drd2*, both showed strong expression in the

AST. *Drd1* was also expressed in the intercalated neurons. *Drd1* expression was mostly in the CeM, detected in >10% of the neurons in multiple clusters (MC-1, MC-11, MC-12, MC-16, and MC-18: seq-c2, seq-c4, seq-c11), whereas *Drd2* was expressed in both CeM (MC-14) and CeC cell types (MC-5). In addition, the opioid receptor kappa 1 (*Oprk1*) was primarily expressed in MC-1 and a subset of BLA neurons (MC-8). The vasoactive intestinal peptide (VIP) receptor 2 (*Vipr2*) was selectively expressed in MC-2, the *Sst* cluster in the CeL/CeC that co-expressed *Crh* and *Tac2*.

## Projections of molecularly defined cell types in the CEA

CEA neurons showed distinct groupings of axonal projection patterns (*Figure 5A*, *Figure 5—figure supplement 1A*, *Figure 5—source data 1–5*). The CEA→BNST projection was most distinct from the other four projection types, with BNST-projecting neurons enriched primarily in the ventral CeM and some at the border between CeM and CeC/L. CEA neurons with descending projections to lateral SN, PBN, PCRt, and vlPAG were largely intermingled. CEA→PBN projecting neurons were the most abundant projection class, with enrichment in CeM and posterior CeC and CeL. The CEA→lateral SN projection followed a similar pattern with retrogradely labeled neurons in anterior CeM and posterior CeM and CeC/L. Neurons from the adjacent AST projected to lateral SN, consistent with previous observation (*Steinberg et al., 2020*). CEA→PCRt neurons primarily originated from the anterior portion of the CeM. CEA→vlPAG neurons were observed in the CeM and the posterior CeL. Neurons co-projecting to PBN and PCRt (n = 132, 6.7%), PBN and lateral SN (n = 76, 3.8%), and PBN and vlPAG (n = 49, 2.5%) were observed at low frequency (*Figure 5—figure supplement 1C*).

A subset of molecularly defined CEA neurons projected to at least one of these five target areas (*Figure 5B*, *Figure 5—figure supplement 1B*). In the CeM, MC-16 (*Nefm/Htr1b/Ebf1/Drd2⁻/Tac2⁻*) was the major cell type that projects to PBN (*Figure 5C*). MC-11 (*Tac1/Sema3c*) was modestly enriched for this projection, and most other CeM cell types showed scattered projections to PBN. In the CeC and CeL, MC-3 (*Sst/Pdyn/Vipr2⁻*) was the major PBN-projecting cell type. In contrast, MC-2, which was adjacent to MC-3 and co-expressed *Sst/Pdyn*, did not show a strong projection to PBN (*Figure 5D*). The different projection patterns of the two transcriptomically similar *Sst* cell types in the CeL may indicate functional differences between these populations. However, both MC-2 and MC-3 neurons project to vlPAG along with a major contribution from CeM cell type, MC-16, and minor contributions from MC-11 (*Sema3c/Tac1*) and MC-13 (*Cyp26b1/Dlk1*). MC-3 and MC-16 clusters were also major lateral SN-projecting cell types. MC-7, a D1 receptor-expressing population primarily in the AST, also showed greater abundance of lateral SN projection compared to the D2-receptor-expressing AST population, MC-20, consistent with canonical direct and indirect striatal pathways to the midbrain (*Gerfen et al., 1990*). MC-3 and MC-16 were also major cell types that project to PCRt. Most neurons with axon collateralizations were from MC-3 or MC-16, with more PBN and lateral SN collaterals observed from MC-3 (61.4%) and more PBN and vlPAG collaterals observed from MC-16 (54.0%) (*Figure 5—figure supplement 1C*). PBN and PCRt collaterals were observed from MC-3, MC-11, and MC-16.

Despite MC-16 being a major CeM descending projection type, neurons in the CeM projecting to BNST were predominantly from MC-13 (*Cyp26b1/Dlk1*) with minor contributions from MC-16, MC-17, and MC-19. MC-6 (*Prkcd*) and MC-10 (*Cyp26b1/Crym⁻*), two CeC/CeL cell types with *Prkcd*-expression, also projected to BNST (*Figure 5—figure supplement 1B–E*). Selected cell clusters and their projections were validated in a third mouse (ANM #3) (*Figure 5—figure supplement 1F*). Consistent with the other two animals, MC-13 (*Cyp26b1/Dlk1*) was the predominant projection type to BNST, and MC-16 (*Nefm/Htr1b/Ebf1/Drd2⁻/Tac2⁻*) was the major CeM projection type to PBN.

## Marker genes predict projection classes

We found that each brain region investigated here received axonal projections originating from multiple CEA neuron types, while only a subset of all cell types in the CEA contributed to these projections. Because neuron types were defined by combinations of marker genes and several genes were expressed by multiple types, we also investigated whether marker genes were predictive of projection targets. For this, we trained a logistic regression classifier that used the marker gene expression in neurons to predict their axonal projections to the five selected brain regions as well as neurons lacking projections to these brain regions (six classes: 'PBN', 'vlPAG', 'PCRt', 'lateral SN', 'BNST', 'unlabeled') (*Figure 6A*). The model had an average area under the receiver-operating characteristic

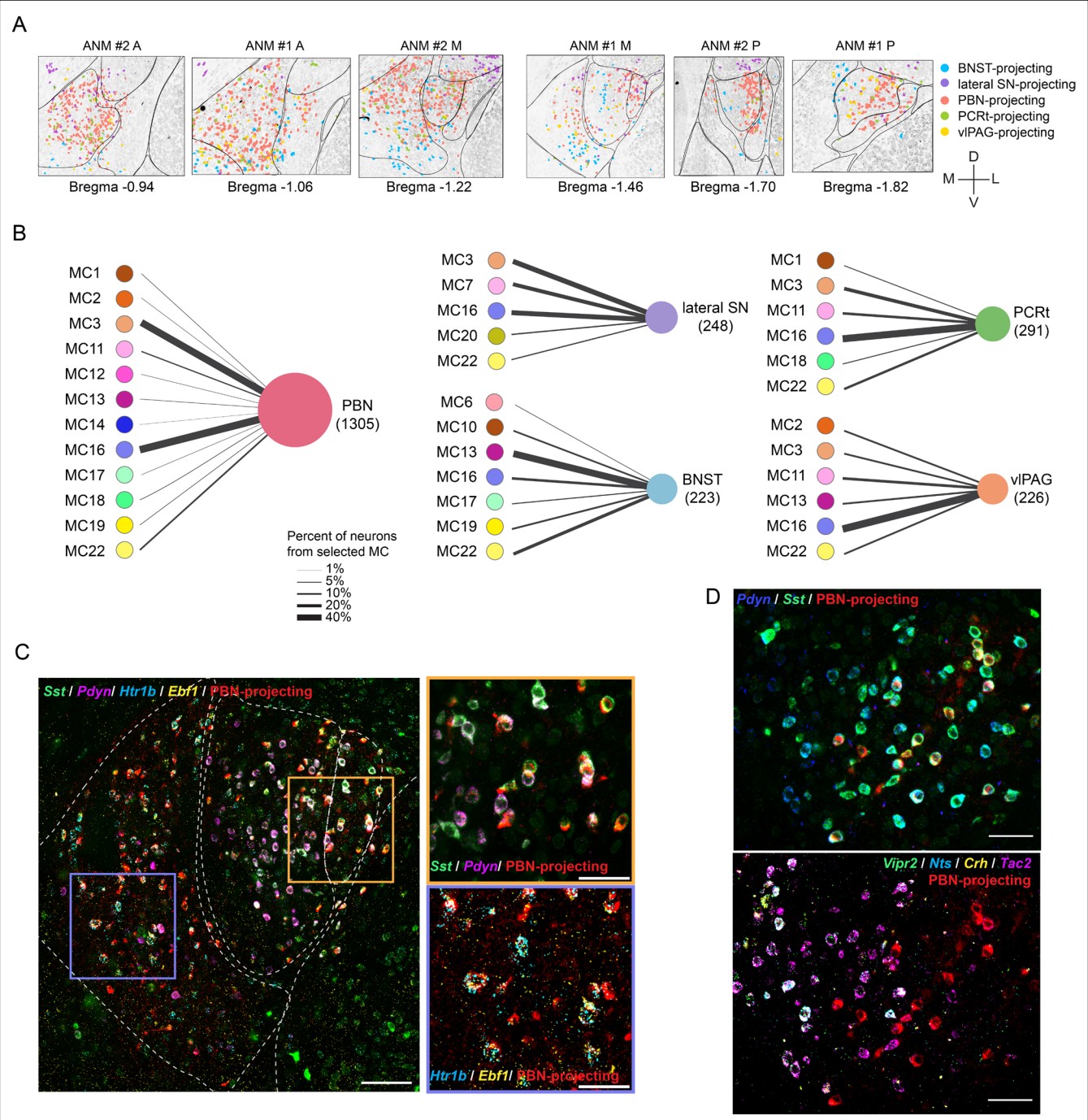

**Figure 5.** Projection of central nucleus of the amygdala (CEA) molecular clusters to five downstream targets. (**A**) Representative images showing neurons projecting to bed nucleus of the stria terminalis (BNST), lateral substantia nigra (SN), parabrachial nucleus (PBN), parvocellular reticular formation (PCRt), and ventrolateral periaqueductal gray (vlPAG). Note that neurons with collaterals (n = 281) are not shown to avoid confusions. Scale bar: 200 μm. (**B**) Molecularly defined CEA neuron types projecting to five downstream brain regions. Line thickness represents percent of neurons from selected MCs projecting to selected brain region. (**C**) Representative image showing the distribution of dominant PBN-projecting clusters, MC-3 and MC-16 and their marker gene expressions. Scale bars: 100 μm (left) and 50 μm (right). (**D**) Representative images showing a subset of *Pdyn/Sst* co-expressing neurons that project to the PBN (top). These neurons are *Vipr2*, *Tac2*, *Nts,* and *Crh*-negative (MC-3) and are localized more laterally (bottom). Scale bars: 50 μm.

The online version of this article includes the following source data and figure supplement(s) for figure 5:

**Source data 1.** Spatial distribution of bed nucleus of the stria terminalis (BNST)-projecting central nucleus of the amygdala (CEA) clusters.

**Source data 2.** Spatial distribution of lateral SN-projecting central nucleus of the amygdala (CEA) clusters.

*Figure 5 continued on next page*

*Figure 5 continued*

**Source data 3.** Spatial distribution of parabrachial nucleus (PBN)-projecting central nucleus of the amygdala (CEA) clusters.

**Source data 4.** Spatial distribution of parvocellular reticular formation (PCRt)-projecting central nucleus of the amygdala (CEA) clusters.

**Source data 5.** Spatial distribution of ventrolateral periaqueductal gray (vlPAG)-projecting central nucleus of the amygdala (CEA) clusters.

**Figure supplement 1.** Molecularly defined neuron types project to different brain regions.

curve (AUC-ROC) of 0.80 (*Figure 6—figure supplement 1A and B*) and f1 score of 0.58 (*Figure 6B*), significantly better in performance compared to data where the relationship between gene expression of individual neurons and their projections was shuffled (p<0.01) (see 'Materials and methods,' *Figure 6—figure supplement 1A*, and *Figure 6B*), indicating high predictive power of marker genes for axon projection targets. The confusion matrix showed that the model had the highest performance predicting BNST (71%) and lateral SN (66%) projections, with more incorrect predictions for hindbrain projections (PBN, PCRt, and vlPAG) (*Figure 6C*). One possibility that could explain lower performance for hindbrain targets was potentially incomplete retrograde labeling of neurons with collateralizations among hindbrain regions, which is indicated by a report of collateralization in axonal reconstructions from rats (*Veinante and Freund-mercier, 2003*). When we trained a model in which the hindbrain projections were grouped (four classes: 'hindbrain', 'lateral SN', 'BNST', 'unlabeled'), the

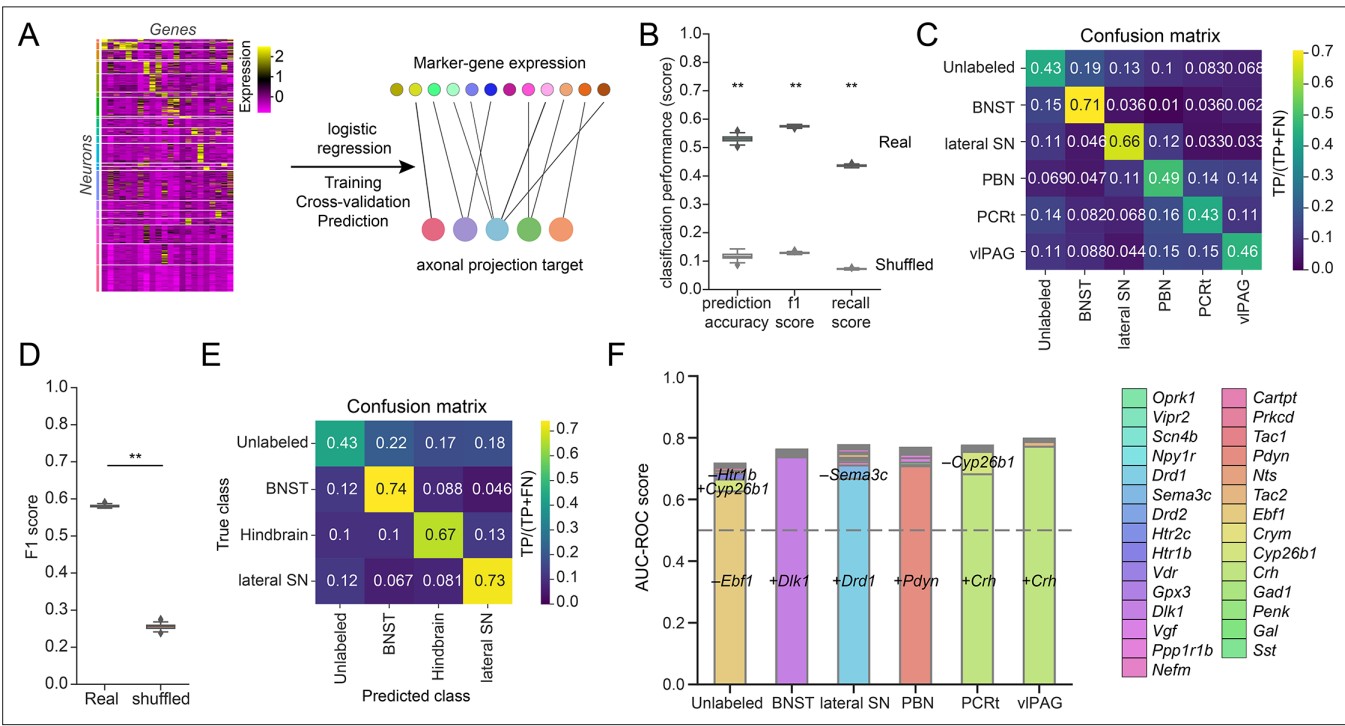

**Figure 6.** Prediction of projection classes with marker genes. (**A**) Schematic of the analysis. Expressions of 29 marker genes from each neuron were used to train a multiclass logistic regression model to predict its axonal projection target. (**B**) Model performance for predicting projection classes (bed nucleus of the stria terminalis [BNST], lateral substantia nigra [SN], parabrachial nucleus [PBN], parvocellular reticular formation [PCRt], ventrolateral periaqueductal gray [vlPAG], and unlabeled) with marker genes, was compared to performance scores generated with shuffled data. p-value was calculated with permutation test. (**C**) Normalized confusion matrix with true class labels in rows and predicted class labels in columns with the logistic regression model in (**B**). Data normalized as true positive (TP) over the total number of neurons from this class (TP + false negative [FN]). (**D**) F1 score for predicting projections to BNST, lateral SN, hindbrain (PBN, vlPAG, PCRt) regions as well as neurons that were unlabeled with marker genes compared to shuffled data. p-value was calculated with permutation test. (**E**) Normalized confusion matrix with true class labels in rows and predicted class labels in columns with the logistic regression model in (**D**). Data normalized as TP over the total number of neurons from this class (TP + FN). (**F**) Area under the receiver-operating characteristic curve (AUC-ROC) scores with sequentially selected features for each projection class. Recursive feature elimination with cross-validation was used to rank features and identify feature(s) that best predict projection targets. Dotted line: AUC-ROC=0.5 Statistics in (**B, D**): permutation test. **p<0.01.

The online version of this article includes the following figure supplement(s) for figure 6:

**Figure supplement 1.** Marker gene expressions predict projection classes.

prediction scores were improved (AUC-ROC score: 0.81, f1 score: 0.59) (*Figure 6D*, *Figure 6—figure supplement 1C*). Consistent with the six-class model, the BNST projection, which had the greatest anatomical separation from the other projection targets, was best predicted by gene expression information (74%). Marker gene expression correctly predicted 73% lateral SN projections and 67% hindbrain projections (*Figure 6E*). The predictions for neurons lacking projections to these regions in both models were lower (43%). Most of the incorrect predictions were false-positive predictions, where unlabeled neurons were predicted to project to three projection classes (BNST: 22%; hindbrain: 17%; and lateral SN: 18%). Among them, 33% of false-positive predictions to BNST were from MC-13 neurons and 35% and 34% of unlabeled neurons predicted to project to hindbrain regions and lateral SN, respectively, were from MC-16 and MC-3. This is potentially related to incomplete retrograde labeling of projection neurons (78% co-labeling efficiency based on CTB and FG co-injection, *Figure 2—figure supplement 1C and D*).

We used the feature coefficients from these models to quantify the relative importance of each gene for predicting the projection types (*Figure 6—figure supplement 1D*) and identified the optimal combination of marker gene(s) to predict neuronal projections based on recursive feature elimination (*Figure 6F*). We found that *Dlk1* had the highest contribution in predicting BNST projection. Expression of *Drd1* and the absence of *Sema3c* expression were predictive of lateral SN projections. *Pdyn*, *Htr1b*, and *Ebf1* were highly predictive of PBN projection. Expression of *Crh* predicted PCRt projections and vlPAG projections, which is consistent with previous studies where *Crh*⁺ neurons were shown to project to lateral PAG (*Fadok et al., 2017*). Although the aforementioned genes best predicted axon projection targets, we found that they can be largely compensated by additional genes (*Figure 6—figure supplement 1E*). We validated the predictive power of *Dlk1*, *Pdyn*, and *Crh* to projection types in a third animal (ANM #3). Consistent with the first two animals, we found that *Dlk1* predicted projections to the BNST, *Pdyn* was predictive of PBN projections, and *Crh* predicted projections to vlPAG (*Figure 6—figure supplement 1F*). Taken together, genes identified in these analyses enriched for CEA neurons projecting to downstream targets, however, they were not fully selective for these projections.

## Discussion

Here, we report a new method for integrating axonal projections with molecular profiling using EASI-FISH in thick tissue samples. This revealed the molecular, spatial, morphological, and connectional diversity in the CEA within a three-dimensional tissue context and uncovered molecularly defined cell types that projected to specific brain regions.

### Technical considerations

Our technical objectives were to extend the EASI-FISH procedure by mapping the location of molecularly defined neuron types using marker genes from scRNA-seq and correlating this with the location of axon projection types in the same samples. We aimed to do this in thick tissue sections because this will eventually facilitate experiments that incorporate EASI-FISH with other experimental modalities such as calcium imaging or electrophysiology. Finally, we configured our experiments so that the automated EASI-FISH analysis pipeline was suitable for the joint analysis of gene expression and retrograde tracer labeling. We opted to develop EASI-FISH for use in conjunction with traditional high-efficiency retrograde tracers, such as CTb and FluoroGold (*Saleeba et al., 2019*). This is because alternative retrograde viral axon tracing tools have considerable tropism for different cell types, and in preliminary experiments, we found poor efficiency of canine adenovirus, AAV2/retro (*Tervo et al., 2016*), and other retrograde viral tools when applied to the CEA projections (*Figure 2—figure supplement 1A and B*). CTb and FG do not survive the tissue digestion step associated with expansion microscopy (ExM), which led us to perform an initial tissue clearing and confocal microscopy step to obtain projection information as well as to develop methods for automatic registration of confocal and SPIM image volumes, which has been challenging in the past. We chose tissue thickness of 100 μm (instead of 300 μm in the previous EASI-FISH study *Wang et al., 2021*) due to the long imaging time and photobleaching associated with confocal imaging as well as compromised image quality due to tissue scattering in thicker tissue sections when using confocal imaging before the expansion microscopy procedure.

Here, we evaluated 22 cell types, using 29 marker genes, for their contribution to 5 projection types, which is more than previous studies for individual samples using retrograde tracers. High-quality neuron somata segmentation provided cell body size and shape information for each neuron, which could be related to neuronal classifications. Our approach to systematically evaluate all CEA cell types for retrograde labeling from five projection targets offers a comprehensive view of the relative contributions of different cell types to each axon projection target. Nevertheless, the absolute number of neurons from each cell type that contribute to each projection is difficult to determine. Although the labeling efficiency of neurons by CTb and FG was high, CEA neurons may arborize in portions of the five target brain regions that were not completely filled with the retrograde tracer by our stereotaxic injection (*Figure 2—figure supplement 2*) and increasing the amount of tracer comes with the tradeoff of overflowing the boundary of the targeted structure. Underfilling the axon-target region may lead to false negatives for classifying neuron contributions to each projection based on measurements of retrogradely labeled neurons, thus underestimating the number of neurons projecting to these regions. Furthermore, this would lead to an undercount of axon collateralization, for which there is prior evidence based on a limited set of single-neuron axon projection reconstructions that have been performed for CEA neurons from rats (*Veinante and Freund-mercier, 2003*). This means that the gene-based predictive model for axon projection target potentially underestimated the predictive power of marker genes due to incomplete retrograde labeling (false-negative labeling). In addition, this study with 5-plex retrograde axon labeling does not include all possible CEA projection targets, such as hypothalamic areas or the nucleus of the solitary tract. Increasing the number of axonal projection targets using this approach will require additional differentially labeled fluorescent retrograde tracers as well as spectral unmixing of the emission of these fluorophores. In this study, we only analyzed samples from male animals, and additional experiments will be needed in female animals to evaluate whether there are differences in molecularly defined cell types or their projections. However, scRNA-seq data from *Peters et al., 2022* using CEA neurons of both male and female animals did not report sex-specific differences. Ultimately, combining EASI-FISH with fluorescent retrograde tracers enables the profiling of tens of thousands of cells for gene expression and projections. In the future, the development of technology for single-neuron axon reconstructions (*Gao et al., 2022*; *Winnubst et al., 2019*) combined with detailed multi-gene expression information would enable evaluation of the complete collateralization of molecularly defined neurons. Overall, our methodology for EASI-FISH combined with fluorescent retrograde axonal tracers is a systematic approach for profiling the gene expression and axon projection targets of single cells in thick tissue volumes.

## Biological insights

The CEA is a major output nucleus of the amygdala, and functional investigation of the CEA has uncovered roles for both defensive and appetitive outputs, which are associated, in some cases, with distinct molecularly defined neuronal populations (*Fadok et al., 2017*; *Yu et al., 2016*). However, it has also been reported that neurons with the same marker gene can be involved in opposite behaviors (*Botta et al., 2015*; *Bowen et al., 2022*; *Cai et al., 2014*; *Fadok et al., 2017*; *Griessner et al., 2021*; *Kim et al., 2017*; *Yu et al., 2016*), raising the possibility that further molecular or anatomical subdivisions exist (*Fadok et al., 2018*; *Moscarello and Penzo, 2022*). We identified molecularly defined cell types in the CEA from scRNA-seq and mapped their detailed spatial locations as well as axonal projections to five major CEA projection targets. This uncovered many new CEA cell types, their participation in CEA circuitry, and revealed their relationships to previously identified CEA marker genes. Recently, single-nucleus RNA sequencing from the CEA in rat identified 13 neuronal types (11/13 neuronal types were from CEA) (*Dilly et al., 2022*). We found good correspondence with major CEA cell types identified by Dilly et al., such as the *Prkcd* and the *Crh* neurons in the CeC and CeL, and the *Drd1* and *Drd2* neurons in the CeM. However, the separation of rat CEA *Sst* subtypes in Dilly et al. was not clear. In addition to these major cell types, our study revealed additional cell-type diversity in the mouse CEA, especially in the CeM, which had been largely underdefined. In addition, our molecularly defined mouse CEA cluster also matched well with two recent studies in mouse CEA (*O'Leary et al., 2022*; *Peters et al., 2022*; *Figure 1—figure supplement 3*).

## Molecular definition of CEA subnuclei molecular boundaries

We found that the primary distinction between molecularly defined CEA cell types corresponded to an anatomical split between CeM and CeC/CeL. This revealed a fundamental distinction between the cellular makeup of these CEA subdivisions, and most cell types mapped specifically to one of these subdomains. Importantly, we used the spatial variation of CEA gene expression to determine combinations of molecular markers to automatically define the boundary of CeL, CeC, and CeM, which have been difficult to determine in the mouse. This provides an approach to increase consistency of anatomical assignments in different studies across animals.

## Molecularly defined CEA neuron types in CeM

We identified many previously undescribed CeM neuron types. For example, MC-16 (*Nefm/Htr-1b/Ebf1/Drd2⁻/Tac2⁻*) is the most abundant CeM cell type but was not previously reported, possibly due to the lack of neuropeptide marker genes. In rats, CeM has been reported to be the primary origin of hindbrain projections (*Cassell et al., 1999*). We found that MC-16 neurons projected primarily to hindbrain and midbrain targets. MC-13 (*Cyp26b1/Dlk1*) is another abundant cell type that is present in both CeM and CeL, and MC-13 projected to the BNST in the forebrain. The different projections of these two cell types are consistent with earlier single-cell anterograde-tracing studies that ignored molecular identity but showed subsets of neurons that had axon projections to either BNST or hindbrain targets (*Veinante and Freund-mercier, 2003*). Moreover, we found that CeM neurons with ascending versus descending projection targets have spatially separate anatomical distributions, with BNST-projecting CeM neurons located more ventrally compared to hindbrain-projecting CeM-projecting neurons.

Although CeM neurons have been reported to be associated with neuropeptide marker genes (*Kim et al., 2017*; *McCullough et al., 2018b*), we found that these neuropeptide-expressing cell types made up only a small proportion of the cells in the CeM. MC-1 is the primary CeM cell type expressing *Tac2*, and we discovered that these cell types have the largest somata in the CEA. *Tac2*-expressing CeM neurons have been previously reported to be associated with appetitive behaviors (*Kim et al., 2017*), but it is challenging to use stereotaxic methods to target these selectively from CeL *Tac2* (MC-2) neurons in mice. We found that this population can be specified by co-expression with *Nefm*, which could potentially be used for the intersectional targeting of this cell type. *Tac1* neurons in the CeM (MC-11) co-expressed *Sema3c*, and this intersection distinguished this population from overlying *Drd1* neurons in the AST that also expressed *Tac1*. In addition, most CeM *Sst* neurons belonged to MC-12, although *Sst* was also found in a subset of *Tac1*-expressing MC-11 neurons. MC-15 was a rare cell type defined by expression of vitamin D receptor (*Vdr*). Although *Vdr* expression has been observed previously in the amygdala (*Liu et al., 2021*; *Stumpf and O'Brien, 1987*) and its expression was shown to be elevated in proestrus rats during the pain response to chicken pox infection (*Hornung et al., 2020*), the functional importance of this *Vdr*-expressing cell type has not been examined. In addition, two other CeM cell types, MC-18 (*Sema3c/Tac1⁻*) and MC-19 (*Gpx3/Gal⁻*), also have not been examined previously. MC-19 was in the ventral CeM and had projections to the BNST, whereas MC-18 had a broader distribution, with projections to PBN and PCRt. Other previously reported CeM markers were expressed in more than one molecularly defined cell type. For example, *Pnoc* (*Hardaway et al., 2019*) was broadly expressed across most major CeM cell types (*Figure 1—figure supplement 2*). While the D2-receptor (*Drd2*) was primarily expressed in two CeM clusters (MC-5 and MC-14, both corresponding to seq-c5 but differing in *Penk* expression), the D1-receptor (*Drd1*) was primarily expressed in the AST cluster (MC-7) and the intercalated cells (MC-21). *Drd1* was also detected (>10%) in multiple CeM clusters (MC-1, MC-11, MC-12, MC-16, and MC-18: seq-c2, seq-c4, seq-c11) (*Figure 1—figure supplement 3*, *Figure 4—figure supplement 4*), largely consistent with a previous report (*Kim et al., 2017*).

## Molecularly defined CEA neuron types in CeL and CeC

The lateral and capsular portions of the CEA contained five primary cell types. *Prkcd* neurons (primarily in MC-6) were the predominant neuron type in the posterior and middle portions of the CeL and CeC. Neurons expressing *Prkcd* have been shown to be suppressed by conditioned stimuli predicting an aversive stimulus (*Haubensak et al., 2010*), to modulate anxiety (*Botta et al., 2015*; *Cai et al., 2014*; *Griessner et al., 2021*), and to suppress food intake associated with either satiety or illness (*Cai et al.,*

*2014*). We identified two populations with *Prkcd* expression, one in rostral CEA (MC-10 corresponds to seq-c6) and one in caudal CEA (MC-6 corresponds to seq-c8, with a higher fraction of neurons expressing *Prkcd*), which likely correspond to the rCEA *Calcrl+* and cCEA *Calcrl+* neurons identified in *Bowen et al., 2022*. A small proportion of neurons in these two populations MC-6 and MC-10 showed projections to BNST, as previously described (*Cai et al., 2014*). *Prkcd*-expressing neurons have been reported to be the primary projections from CeC/CeL→BNST (*Ye and Veinante, 2019*); however, we found that this marker gene is associated with a small proportion (15.2%) of CeC/CeL neurons that project to BNST. Instead, most BNST-projecting neurons are from MC-13 (31%), which straddles the boundary between CeL and CeM, and the second-most abundant BNST-projecting population was MC-10 (8.1%). We did not evaluate intra-CeA connectivity, but *Prkcd* neurons of the CeC and CeL have been reported to also form connections with CeM neurons (*Ye and Veinante, 2019*), some of which go on to project to the PAG (*Haubensak et al., 2010*). In addition, *Prkcd* neurons are also engaged in a recursive inhibitory local circuit with CeL *Sst* neurons (*Fadok et al., 2017*; *Hunt et al., 2017*).

Consistent with previous reports, we detected significant levels of co-localization in CeC/CeL of previously used marker genes such as *Sst*, *Crh*, *Nts,* and *Tac2*, which were distinct from *Prkcd*-expressing neurons (*Kim et al., 2017*; *McCullough et al., 2018b*; *Ye and Veinante, 2019*). Yet, we identified separate CeC/CeL *Sst*-expressing subpopulations that can be selectively targeted based on the co-expression of previously unknown molecular markers. MC-3 strongly expressed *Sst* and was spatially intermingled with *Prkcd* neurons. A closely related *Sst*-expressing cell type, MC-2, expressed *Vipr2*, was partially offset to the medial portion of the CeL and co-expressed varying proportions of *Sst, Nts, Crh* (*Kim et al., 2017*; *McCullough et al., 2018b*). MC-2 was restricted to posterior and middle CEA sections, while MC-3 was also in anterior CeC. These *Sst*-expressing clusters might correspond to distinct functional subpopulations and could explain seemingly contradictory reports on the behavioral role of SST and CRH neurons in the CEA (*Fadok et al., 2017*; *Kim et al., 2017*; *Yu et al., 2016*). Indeed, *Sst*-expressing neurons from the CeL have been reported to be the primary projection to PBN, and we found that it was the MC-3 population that was the major descending projection from CeC/CeL to the PBN, PCRt, and lateral SN. Both MC-3 and MC-2 neurons participate in projections to vlPAG. The functional significance of two *Sst* neuron populations in CeL is not known, but functional activation of CeL$^{CRH}$ neurons, possibly projecting to vlPAG (likely MC-2), has been shown to promote flight responses (*Andero et al., 2014*), whereas activation of CeL$^{SST}$ projecting to vlPAG (likely MC-2 and MC-3) promotes freezing (*Fadok et al., 2017*; *Kim et al., 2017*; *Yu et al., 2016*). However, these earlier studies may include contributions from neurons expressing these genes in CeM, thus state-dependent effects (*Fadok et al., 2018*; *Moscarello and Penzo, 2022*) that primarily affect separate CEA subpopulations cannot be excluded.

In the anterior CeC/L, the predominant cell type, MC-10 (*Cyp26b1*/*Crym$^-$*), has not been reported previously and, as mentioned above, is the primary CeC/L projection to BNST. The anterior CeC also contained a *Drd2* cell type co-expressing *Scn4b* (MC-5) that extended ventrally from the AST and projected to lateral SN, which likely corresponded to previously reported *Drd2* neurons in and around CEA that enhance conditioned freezing (*McCullough et al., 2018a*). However, *Drd2* expression is not limited to MC-5 (e.g., MC-14 in CeM), and we anticipate that the use of intersectional targeting with *Drd2/Scn4b* will facilitate selective targeting of this CeC cell type. A previous study reported that most *Prkcd*-expressing neurons co-expressed *Drd2* and that most CeC/L *Drd2* was expressed in these neurons using a *Drd2::EGFP* transgenic mouse (*De Bundel et al., 2016*). In contrast, we find that direct detection of RNAs by scRNA-seq as well as high-sensitivity EASI-FISH identified *Drd2* expression in only a proportion of *Prkcd*-expressing neurons (7.7% scRNA-seq, 43.0% FISH).

## Long-range projections of CEA molecular cell types

We observed a complex set of relationships in the long-range neuronal projection network of the CEA, where target regions receive inputs from multiple molecularly defined cell types, with molecularly defined cell types projecting to more than one target region. We also observed distinct projection patterns from closely related cell types. For example, the MC-3 *Sst*-expressing cell type in the CeL projects to multiple descending brain areas but the closely related and anatomically intermingled MC-2 (*Sst/Vipr/Nts/Tac2* co-expressing) cell type had an ~11-fold lower proportion of retrogradely labeled neurons. The most abundant projections from CEA were to PBN and involved 11 (50% of

total) molecularly defined cell types in the CEA, while MC-16 from CeM and MC-3 from CeC/CeL were the primary cell types projecting to lateral SN, PBN, and PCRt. Thus, dual control of CEA projection targets by CeL and CeM is a common organizational characteristic of CEA output. However, it remains to be investigated whether these distinct molecular clusters with common projection targets serve specific functions, such as associative learning to salient stimuli, feeding or predatory behavior that have been previously linked to CEA projections to the lateral SN, PBN, or PCRt, respectively (*Douglass et al., 2017*; *Han et al., 2017*; *Steinberg et al., 2020*). Of note, two of the CeM clusters with distinct projection patterns identified in this study likely corresponded to the novel CeM cell types identified in *O'Leary et al., 2022*. MC-16 that projects to multiple hindbrain targets corresponds to *Isl1*$^+$, while MC-13 that showed a spatial distribution between CeL and CeM likely corresponds to *Nr2f2*$^+$ neurons. And we show in this study that MC-13 is a major BNST-projecting population. In addition to these findings, our study offers a more comprehensive view of the co-projection relationships of all CEA clusters in CEA subnuclei that is not limited by retrograde viral tropism.

We also evaluated the predictive accuracy of marker genes for individual projection types. Because multiple cell types contributed to each projection type, it is important to determine whether marker genes expressed across multiple cell types would be associated with an axon projection. Overall predictive scores (AUROC) ranging from 0.78 to 0.83 indicate good predictability. When we examined the predictive power of top-ranked marker genes, we found, on average, that single marker genes would be associated with ~70%/30% split between true positives and false positives for a given projection type (~80%/20% for *Crh* prediction of vlPAG). However, the prediction scores for these marker genes may be underestimated because of false-negative neurons that projected to portions of the targeted regions not contacted by the tracer injections and were thus unlabeled by tracers. Nevertheless, applications involving selective transgene expression in molecularly distinct neuronal projection types are likely best achieved using retrograde viral approaches, possibly in conjunction with marker genes that represent the distinct cell types contributing to that projection.

Together, our findings define the molecular neuronal subtypes of the CEA, reveal major differences in the molecular cytoarchitecture of CeC, CeL, and CeM, and relate molecular subtypes to major projection targets. Future work will be required to establish conditions for targeting these cell types in mice with transgenes. In addition, previous studies indicate that local CEA circuitry is highly organized (*Fadok et al., 2017*; *Haubensak et al., 2010*; *Hunt et al., 2017*; *Li et al., 2013*). It remains to be investigated to which extent molecular identity predicts local connectivity within and across CEA subnuclei. This study will provide a basis to use marker genes to facilitate experiments that address whether distinct molecular CEA cell types, in conjunction with their projection targets, control distinct motivated appetitive and aversive behaviors and whether CEA output pathways convey more abstract, scalable, and state-dependent information.

# Materials and methods

## Key resources table

| Reagent type (species) or resource | Designation | Source or reference | Identifiers | Additional information |
|---|---|---|---|---|
| Peptide, recombinant protein | Cholera Toxin Subunit B (Alexa Fluor-488) | Thermo Fisher | Cat. # C34775 | |
| Peptide, recombinant protein | Cholera Toxin Subunit B (Alexa Fluor-555) | Thermo Fisher | Cat. # C34776 | |
| Peptide, recombinant protein | Cholera Toxin Subunit B (Alexa Fluor-594) | Thermo Fisher | Cat. # C34777 | |
| Peptide, recombinant protein | Cholera Toxin Subunit B (Alexa Fluor-647) | Thermo Fisher | Cat. # C34778 | |
| Chemical compound, drug | FluoroGold | Fluorochrome | | |
| Chemical compound, drug | Melphalan | Cayman Chemicals | Cat. # 16665 | |
| Chemical compound, drug | Acryloyl-X, SE | Thermo Fisher | Cat. # A20770 | |
| Commercial assay or kit | RNase-Free DNase Set | QIAGEN | Cat. # 79254 | |

*Continued on next page*

*Continued*

| Reagent type (species) or resource | Designation | Source or reference | Identifiers | Additional information |
|---|---|---|---|---|
| Commercial assay or kit | Proteinase K | NEB | Cat. # P8107S | |
| Chemical compound, drug | DAPI | Sigma | Cat. # D9542 | |
| Chemical compound, drug | Janelia Fluor 669, SE | Tocris | Cat. # 6420 | |
| Chemical compound, drug | N,N,N',N'-Tetramethyl ethylenediamine | Sigma | Cat. # T22500 | |
| Chemical compound, drug | Ammonium persulfate | Sigma | Cat. # A3678 | |
| Chemical compound, drug | Acrylamide solution | Sigma | Cat. # A4058 | |
| Chemical compound, drug | 4-Hydroxy-TEMPO | Sigma | Cat. # 176141 | |
| Chemical compound, drug | N, N'-Methylenebisacrylamide | Sigma | Cat. # M7279 | |
| Chemical compound, drug | Acrylamide | Sigma | Cat. # A9099 | |
| Chemical compound, drug | Acrylic acid | Sigma | Cat. # 147230 | |
| Chemical compound, drug | DMSO | Sigma | Cat. # 570672 | |
| Chemical compound, drug | MOPS buffer | Sigma | Cat. # M1254 | |
| Chemical compound, drug | 20× SSC | Thermo Fisher | Cat. # AM9763 | |
| Chemical compound, drug | Nuclease-free water | Thermo Fisher | Cat. # AM9932 | |
| Chemical compound, drug | NaOH | Fisher scientific | Cat. # SS267 | |
| Chemical compound, drug | Poly-L-lysine | Pelco | Cat. # 18026 | |
| Chemical compound, drug | Dextran sulfate 50%, 100ML | Sigma | Cat. # S4030 | |
| Chemical compound, drug | Formamide | Fisher Scientific | Cat. # BP227-100 | |
| Chemical compound, drug | PBS | Fisher Scientific | Cat. # BP24384 | |
| Chemical compound, drug | RNase away/DNase away | Fisher Scientific | Cat. # 10328011 | |
| Chemical compound, drug | Photo-Flo 200 | EMS | Cat. # 74257 | |
| Commercial assay or kit | QIAquick Nucleotide Removal Kit | QIAGEN | Cat. # 28304 | |
| Strain, strain background (mouse, male) | C57Bl/6 | Jackson Laboratory | JAX stock #000664 | |
| Sequence-based reagent | HCR probes | Molecular Instrument | N/A | |
| Sequence-based reagent | HCR Amplifier B1 | Molecular Instrument | N/A | |
| Sequence-based reagent | HCR Amplifier B2 | Molecular Instrument | N/A | |
| Sequence-based reagent | HCR Amplifier B3 | Molecular Instrument | N/A | |
| Sequence-based reagent | HCR Amplifier B4 | Molecular Instrument | N/A | |
| Sequence-based reagent | HCR Amplifier B5 | Molecular Instrument | N/A | |
| Sequence-based reagent | Custom-DNA probe | This study | Ribosomal RNA probes | Sequence: gcgggtcgccac gtctgatctgaggtcgcg/3 ATTO550N/ |
| Software, algorithm | EASI-FISH pipeline | *Wang et al., 2021* | | https://github.com/JaneliaSciComp/multifish |
| Software, algorithm | Seurat 4.0.1 | *Stuart et al., 2019* | RRID:SCR_016341 | https://satijalab.org/ |
| Software, algorithm | Fiji | ImageJ | https://imagej.net/software/fiji/ | |

*Continued on next page*

*Continued*

| Reagent type (species) or resource | Designation | Source or reference | Identifiers | Additional information |
|---|---|---|---|---|
| Software, algorithm | Python v3.7 | | RRID:SCR 008394 | https://www.python.org/ |
| Software, algorithm | n5-viewer | *Saalfeld et al., 2022*; Saalfeld lab | | https://github.com/ saalfeldlab/n5-viewer |
| Software, algorithm | Napari | *Napari contributors, 2019* | | https://napari.org/ |
| Other | Zeiss Lightsheet Z.1 microscope | Zeiss | | https://www.zeiss.com/ microscopy/us/products/ imaging-systems/ light-sheet-microscope- for-lsfm-imaging-of-live- and-cleared-samples- lightsheet-7.html |
| Commercial assay or kit | Press-to-Seal Silicone Isolator with Adhesive | Thermo Fisher | Cat. # P24743 | |
| Commercial assay or kit | 8 mm glass coverslip | Harvard Apparatus | Cat. # BS4 64-0701 | |
| Other | Zeiss Lightsheet Z.1 imaging holder | Svoboda Lab and Janelia Experimental Technology | | The design of the imaging holder can be found at https://www.janelia. org/open-science/zeiss- lightsheet-z1-sample- holder |
| Other | CEA scRNA-Seq | This study | GEO: GSE213828 | Sequencing data included in this study is available through GEO: https://www. ncbi.nlm.nih.gov/geo/ |
| Other | CEA EASI-FISH data | This study | | FISH and projection data included in this study are publicly available through the following links: https:// doi.org/10.25378/janelia. 21171373; http://multifish- data.janelia.org/ |

## Animal model and subject details

Adult C57Bl/6J male mice (8 weeks old at the beginning of the experiments) were used. All methods for animal care and use were conducted according to National Institutes of Health guidelines for animal research and approved by the Institutional Animal Care and Use Committee (IACUC) at Janelia Research Campus (Protocol number: 18-174). Mice were housed in a 12 hr light/12 hr dark cycle and had ad libitum access to water and chow diet.

## Single-cell RNA sequencing

Single-cell RNA sequencing was focused on the central amygdala (CEA). For visually guided dissection of the CEA, we used the fluorescent boundaries of the robust PBN→CEA axon projection. This has the advantage of marking the CEA without directly expressing a transgene in neurons to be analyzed by scRNA-seq. For this, male C57Bl/6J mice were bilaterally injected with AAV2/1-CAG-GFP (Capsid from AAV1 and ITR from AAV2) (titer 5.5E+12 vg/ml; Janelia Viral Tools facility) into the parabrachial nucleus (PBN; 100 nl per hemisphere, coordinates from bregma: anterior-posterior [AP] –5.2 mm, medial-lateral [ML] 1.15 mm, dorsal-ventral [DV] 3.25 mm). For AAV injection under stereotaxic control, mice were anesthetized using isoflurane (3–5% for induction, 1–2% for maintenance) in oxygen and fixed on a stereotactic frame (Model 1900, Kopf Instruments). Injections of buprenorphine (0.1 mg per kg body weight subcutaneously before anesthesia) and ketoprofen (5 mg per kg body weight after the surgery and every 24 hr for 2 days postoperatively) were provided for analgesia. Ophthalmic ointment was applied to avoid eye drying. The body temperature of the animal was maintained at 36°C using a feedback-controlled heating pad. A pulled glass pipette (tip diameter ~20 μm) was connected to a microinjection system (Oil Microinjector, Narishige) and lowered into the brain at

the desired coordinates with the stereotaxic micro-positioner (Model 1940, Kopf Instruments). After a waiting time of 10 min, the pipette was slowly removed, and the wound was closed with a surgical suture.

Four weeks after PBN injection, mice were sacrificed to collect CEA neurons for scRNA-seq. The manual sorting procedure to isolate cell bodies from micro-dissected brain slices was similar to a previously described protocol (*Hempel et al., 2007*). Briefly, mice were deeply anesthetized with isoflurane and decapitated to collect 300 µm coronal brain slices. The CEA was manually dissected with spring scissors using the GFP signal as guidance. Afferent fibers from the PBN specifically labeled the CEA but not surrounding nuclei such as the basolateral or medial amygdala, or the amygdala-striatal transition zone, and thus allowed for specific dissection of the CEA without expression of the fluorescent protein in CEA neurons (*Figure 1—figure supplement 1B-C*). Two to three tissue sections from each animal were taken to cover the entire CEA and subjected to protease digestion, after which cells were dissociated. Intact neurons were manually selected into individual wells. Neurons collected from seven animals were pooled for sequencing. Sorted single cells were lysed with 3 µl lysis buffer (0.2% Triton X-100 [Sigma] and 0.1 U/µl RNase inhibitor [Lucigen]) and cDNAs were prepared using the Smart-SCRB chemistry as described previously (*Cembrowski et al., 2018*; *Xu et al., 2020*). Barcoded cDNAs were then pooled to make cDNA libraries, and the cDNA libraries were sequenced on a NextSeq 550 high-output flowcell with 26 bp in read 1 to obtain the barcode and UMI, and 50 bp in read 2 for cDNA. PhiX control library (Illumina) was spiked in at a final concentration of 15% to improve color balance in read 1. Libraries were sequenced to an average depth of 6,611,566 ± 92,440 (mean ± S.D.) reads per cell.

Sequencing alignment was performed similar to a previous report (*Gur et al., 2020*). Sequencing adapters were trimmed from the sequencing reads with Cutadapt v2.10 (*Martin, 2011*) prior to alignment with STAR v2.7.5c (*Dobin et al., 2013*) to the *M. musculus* GRCm38.90. genome assembly from Ensembl (https://asia.ensembl.org/index.html). Gene counts were generated using the STARsolo algorithm (https://github.com/alexdobin/STAR/blob/master/docs/STARsolo.md). Gene counts for the subset of barcodes used in each library were extracted using custom R scripts.

## scRNA-seq analysis

First, genes that were expressed in <5 cells and cells with <200 detected genes were removed from the dataset. Cell doublets/multiplets and low-quality cells were filtered based on the total number of detected genes (1500-7500), relative abundance of mitochondrial transcripts (percent.mito < 0.055) and number of unique molecular identifiers (nUMI) per cell ($<2 \times 10^5$), respectively. The resulting dataset consisted of 1,626 cells and 33,372 genes. Next, gene expression in remaining cells were normalized to total expression and log transformed. The top 5,000 highly variable features were selected after variance-stabilizing transformation (vst) (*Hafemeister and Satija, 2019*). Gene expression was then z-score transformed after regressing out the effects of latent variables including nUMI and percent.mito. PCA was performed on z-score normalized data. The top 40 PCs from the PCA were used for clustering analysis. Non-neuronal clusters were identified by expression of non-neuronal markers (e.g., *Aqp4, Olig1, Olig2, Opalin, Pdgfra, Ctss, Flt1, Epas1, Esam, Krt18, Jchain, Pecam1*) and absence of neuronal markers (*Snap25, Syp, Tubb3, Map1b, Elavl2, Gad1, Gad2*, etc.) and removed. Neuronal clusters outside of the CEA (based on expression of *Neurod6, Slc17a7, Slc17a6, Lhx8,* and *Lhx6*) were also removed from subsequent analysis. The resulting dataset consisted of 1,393 cells and 33,372 genes.

Next, as described above, variable features were identified and used for dimensionality reduction and clustering analysis. The shared nearest neighbor (SNN) with modularity optimization (Louvain algorithm with multilevel refinement procedure and 10 iterations) clustering algorithm implemented in Seurat was used to identify cell clusters. Silhouette score and the Jaccard index distribution after bootstrapping were used to determine the optimal resolution and neighborhood size for clustering, as described previously (*Wang et al., 2021*). The Silhouette score is a measure of how similar a cell is to its own cluster compared to other clusters, and the silhouette score for each cell in a specified cluster was calculated as the Euclidean distance in PCA space using the CalculateSilhouette function in Seurat. For bootstrap analysis, we randomly selected 80% of cells from the integrated dataset and performed dimensionality reduction and clustering. We then calculated the Jaccard similarity index between the most similar new cluster and the original cluster. This procedure was repeated 100 times,

and the distribution of Jaccard similarity index across clusters was plotted and used to evaluate cluster stability. Clusters with high stability have consistently high Jaccard similarity index with bootstrapping. This bootstrap analysis was performed using *sclusteval* package in R (*Tang et al., 2020*) with modifications. Based on these evaluations (*Figure 1—figure supplement 1E*), the following parameters were chosen for graph-based clustering of the CEA neurons: k.param=25, resolution = 1.5. Clustering analysis identified 13 neuronal clusters, whose identities were assigned based on expression of enriched genes. BuildClusterTree() function in Seurat was used to generate the dendrogram representing transcriptomic relationships of neuronal clusters. It constructs a phylogenetic tree based on a distance matrix constructed in the PCA space on averaged cell from each cluster. For data visualization, the top 30 PCs were used to calculate the UMAP, with n.neighbors=30L and min.dist=0.3. Differential gene expression analysis was performed using the FindAllMarkers function in Seurat (Wilcoxon rank-sum test, logfc.threshold=0.75, min.pct=0.25), with p-values adjusted based on the Bonferroni correction. The top 50 enriched genes for each neuronal cluster are provided in *Supplementary file 1*.

## Marker gene selection for EASI-FISH

To identify marker genes for EASI-FISH, we first started with the differentially expressed genes outlined above . We applied a series of selection criteria designed to allow classification of a maximum number of unique cell types using the fewest number of genes possible. As such, in addition to limiting our search to genes with an adjusted p-value cutoff of at least 0.05 and an average log-fold change of 0.55 or over, we also specifically selected markers with as close to binary 'on/off' expression patterns in the cell type of interest as possible, based on high percentage of marker positive cells in the target population compared to low percentage of marker positive cells outside the target population (displayed as pct.1 and pct.2 in *Supplementary file 1*, respectively). This provided us with a candidate gene list, which we cross-validated with Allen ISH data (*Lein et al., 2007*) based on marker gene expression in the CEA. To independently validate marker gene selection, we also built a combinatorial marker gene panel with a greedy algorithm implemented in *mfishtools*, where marker genes were selected one at a time to maximize the fraction of neurons that can be correctly assigned to the correct identity.

In addition to marker genes, differentially expressed GPCRs (e.g., *Npy1r*, *Drd1*, *Drd2*, *Htr1b*, and *Htr2c*) were included because of their potential interest as neuromodulatory receptors. A total of 29 marker genes were chosen, and we note that this is not the only combination of genes that could feasibly serve to represent these molecularly defined cell types. Based on cross-correlation analysis, we were able to correctly assign cluster identity to around 60% of neurons in the scRNA-seq dataset based on selected marker genes. Addition of another 71 marker genes (total of 100) provided only modest improvement in the assignment accuracy to 65%. With all genes in the dataset, the cross-correlation analysis assigned 87.8% of neurons to their original identity, with the lowest assignment accuracy in seq-c13 (70%).

## Integration of scRNA-seq datasets

We compared our scRNA-seq data with a recently published scRNA-seq dataset in the CEA (*O'Leary et al., 2022*) by integrating the two datasets using the canonical-correlation analysis (CCA) implemented in Seurat (v4.2.0). Processed gene count expression and metadata matrices for the O'Leary et al. dataset were directly downloaded from Gene Expression Omnibus (GEO). First of all, gene expressions were normalized to total expression and then log-transformed. Then the top 5,000 highly variable features were selected from each dataset after variance-stabilizing transformation (vst) (*Hafemeister and Satija, 2019*) and used to identify integration anchors through the FindIntegrationAnchors function in Seurat (parameters used: dims=1:50, k.score=30, reduction='cca', anchor. features=5,000). 3,118 anchors were selected and used to compute a weighted integration vector (k. weight = 100), which was then used to transform the two datasets into a common space (merging O'Leary et al. dataset into our dataset) using Seurat's IntegrateData function. The integrated dataset with a total of 2,222 cells was then used as input for dimensionality reduction and clustering. UMAP was computed and used for visualization with the following parameters (dims=1:30, n.neighbors=30L, min.dist=0.30, n.epochs=500, seed.use = 6). SNN based clustering was performed on integrated data with the following parameters: k.param=25, resolution = 1.0. To compare scRNA-seq clusters between the integrated data and our study in *Figure 1*, Jaccard similarity index was computed using sclusteval package in R (*Tang et al., 2020*) with modifications.

## Projection-FISH method

### Retrograde labeling

C57Bl/6J male mice (8 weeks old) were used for all EASI-FISH experiments with retrograde tracer labeling. The non-toxic retrograde tracers cholera toxin b (CTB) conjugated with different fluorophores (Alexa Fluor-488, Alexa Fluor-555, Alexa Fluor-594, Alexa Fluor-647; all Thermo Fisher, 0.5%) and FluoroGold (FG; Fluorochorome, 2%) were injected into the left hemisphere of five selected projection areas of the CEA: the bed nucleus of the stria terminalis (BNST; coordinates from bregma: AP 0.25 mm, ML 1.0 mm, DV 4.4 mm), the lateral part of the substantia nigra (lateral SN; AP –3.65 mm, ML 1.8 mm, DV 3.8 mm), the ventrolateral PAG (vlPAG; AP –4.65 mm, ML 0.5 mm, DV 2.35 mm), the parabrachial nucleus (PBN; AP –5.2 mm, ML 1.15 mm, DV 3.25 mm), and the parvocellular reticular nucleus (PCRt; AP –6.4 mm, ML 1.25 mm, DV 4.7 mm). The surgery was performed as described above. Animals received up to 0.5 ml 0.9% saline/0.5 ml 5% glucose (subcutaneously) during the surgery. 50 nl of retrograde tracer was injected into each region. For animal #1, BNST was injected with FG, lateral SN CTB-647, vlPAG CTB-594, PBN CTB-555, and PCRt CTB-488. For animal #2, BNST was injected with FG, lateral SN CTB-647, vlPAG CTB-488, PBN CTB-555, and PCRt CTB-594. For animal #3, BNST was injected with FG, lateral SN CTB-594, vlPAG CTB-647, PBN CTB-555, and PCRt CTB-488.

### Tissue fixation and preparation

8–10 weeks after CTB and FG injection, animals were anesthetized with isoflurane and perfused with RNase-free PBS (15 ml) followed by ice-cold 4% paraformaldehyde (PFA) (50 ml). Brain tissue was dissected and fixed in 4% PFA overnight before sectioning on a vibratome. Brain coronal slices (100 μm) were sectioned and stored in 70% ethanol (to preserve RNA) at 4°C for up to 6 months. The CEA region (approximately –0.7 to –1.9 mm AP from bregma) was cut out using anatomical landmarks as boundaries. Three sections per animal were analyzed, an anterior section (located between –0.7 and –1.0 mm AP from bregma), a medial section (–1.1 to –1.5 mm), and a posterior section (–1.6 to –1.9 mm). For ease of orientation and optimal imaging, the tissue was cut as a rectangle (~2.5 × 4 mm). An RNase-free paintbrush was used for tissue handling.

### Tissue clearing and projection class imaging

The tissue slice was rehydrated in PBS at room temperature (RT) (2 × 15 min) and incubated in PBS with 8% SDS (Fisher Scientific) at RT (4 hr) for tissue clearing. Cleared tissue was rinsed in 2× SSC (2 × 1 hr) and stained in PBS with 200 ng/ml DAPI (30 min), followed by rinsing in PBS (2 × 30 min). Next, the sample was transferred to a glass-bottom 6-well plate and mounted with index-matched imaging medium (60% sucrose in PBS) and cover-slipped. The Zeiss 880 confocal with Plan-Apochromat 20×/0.8 M27 objective was used to collect CTB, FG, and nuclear DAPI fluorescence signals. The following four image tracks (z-stack) were collected sequentially. Track 1: CTB-594 was excited by 594 nm laser and signals in the range of 597–633 nm were collected. Track 2: CTB-555 and CTB-647 were excited simultaneously with 561 nm and 633 nm laser and signals in the range of 562–597 nm were collected for CTB-555 and 650–690 nm for CTB-647. Track 3: CTB-488 was excited by 488 nm laser and signal detected in the range of 500–571 nm. Track 4: nuclear DAPI and FG were excited with 405 nm laser, with DAPI signal collected in the range of 410–480 nm and FG in the range of 500–695 nm. Single-color tissue samples were used and imaged separately to correct for signal crosstalk post hoc using Fiji spectral unmixing plugin. Tiled fluorescent images were taken with 10% overlap between tiles, with pixel size 0.73 μm in x and y dimension, and z step size of 0.77 μm.

To determine whether the tissue clearing and imaging procedure compromise RNA quality, genes uniquely expressed in selected neuronal subset with known expression levels according to scRNA-seq data (*Ezr* in *Prkcd*+ neurons and *Igf1* in *Pmch*+ neurons) were used to determine RNA quality with EASI-FISH. Transcript spot counts in conditions with and without 8% SDS clearing and confocal imaging were assessed and compared.

### EASI-FISH procedure modifications

After confocal imaging, the tissue sample was recovered and rinsed in PBS (2 × 15 min) and incubated in MOPS buffer (20 mM, pH 7.7, 30 min). Tissue was incubated overnight (37°C) in MOPS buffer (50 μl)

with 1 mg/ml MelphaX and gelled according to the EASI-FISH protocol (*Wang et al., 2021*). Proteinase K digestion in EASI-FISH procedure removes most proteins, including retrograde tracers, and frees up all fluorescence channels. To register confocal image volumes with EASI-FISH images from light-sheet fluorescence imaging, round 0 imaging was performed before DNase I digestion. Fluorescence in situ hybridization with a ribosomal RNA probe produced a cytosolic stain and was used to register to cytoDAPI in EASI-FISH rounds and nuclear DAPI staining was applied to allow registration to confocal imaging. Specifically, tissue-gel sample was first equilibrated in hybridization buffer (500 µl) for 30 min at 37°C. Samples were then hybridized with a ribosomal RNA probe (sequence: gcgggtcgccacgtctgatc tgaggtcgcg/3ATTO550N/) (1 µM) in hybridization buffer (300 µl) overnight at 37°C. The next morning, samples were washed in probe wash buffer (2 × 30 min), followed by PBS (2 × 30 min) at 37°C. The tissue-gel sample was then stained in PBS with 200 ng/ml DAPI (30 min), followed by rinsing in PBS (2 × 30 min), and imaged on a Zeiss Z.1 Lightsheet microscope. After imaging, the tissue-gel sample was DNase I digested and processed according to the EASI-FISH protocol (*Wang et al., 2021*).

## Projection-FISH data analysis

### EASI-FISH data processing

EASI-FISH data analyses, including image stitching, registration, segmentation, spot detection, and assignment, were performed using the pipeline described before (*Wang et al., 2021*). For determining the number of cells expressing a gene, we first calculated the number of background spots in samples lacking that gene to account for nonspecific spots in tissue samples, which gives on average 10 background spots/cell. Based on this, we set the threshold to define positive cells as any cell with a spot count greater than 10.

### Projection data analysis

First, projection images were aligned to EASI-FISH images using nuclear DAPI signals. Nuclear DAPI signal from projection image taken by confocal microscope (moving) was used to register to nuclear DAPI signal in round 0 EASI-FISH image (fixed) using Bigstream (*Fleishman, 2021*) (https://github.com/GFleishman/bigstream) with modifications. First, 50 features on average were manually selected from both images, matched with RANSAC to calculate the global affine transform for moving image. After applying this global affine transformation, the transformed image volumes were split into overlapping chunks for further processing as described before. Another round of feature selection using Laplacian of Gaussian (LoG) filter and RANSAC-based affine transformation was performed on image chunks, followed by deformable registration. All registration steps on image chunks were executed in parallel. The global affine, piecewise affine, and piecewise deformable transforms were composed to a single displacement vector field stored in N5 format. The forward transform was applied to all projection channels to align them to EASI-FISH images (fixed).

The EASI-FISH segmentation mask was applied to registered, crosstalk corrected confocal images to calculate the average fluorescence intensity of CTB and FG signals in each neuron (ROI). The fluorescence intensities from each neuron were thresholded based on the receiver-operating characteristic (ROC) curve to determine projection types. Briefly, signal intensities were z-score normalized for each projection. Neurons were then randomly sampled (100 neurons from each projection at each selected normalized threshold) and manually inspected at different fluorescence intensity cut point to generate the ROC curve. The Youden index (J) was used to determine the optimal threshold for each projection, and neurons with fluorescence intensity above this threshold were classified as targetX-projecting and neurons below this threshold were classified as unlabeled.

### FISH clustering analysis

Cells were first clustered after PCA using graph-based SNN clustering method implemented in Seurat (4.0.1) with Louvain algorithm with multilevel refinement to remove non-neurons as well as neurons outside of the CEA. Parameters for clustering were determined with bootstrap analysis by subsampling 80% of the data for 100 times (k params: 25 and resolution: 0.2). The non-neurons were identified based on small cell body volume and lack of neuronal marker gene expression and removed. 33,139 out of 42,619 cells from 6 samples (*Supplementary file 3*) were used for downstream analysis. As described before, the clustering analysis was performed after PCA of z-score normalized spot counts (without logarithmic transformation to minimize the weight of false-positive spot detections).

Subsequent clustering was performed on CEA neurons with parameters optimized with boot-strapping analysis to maximize the number of stable clusters (k parameter: 45; resolution: 1.0). Two confusing clusters were merged, and the clusters were reordered (from high to low) based on the cluster-average total spot counts. For visualization, normalized expression of 29 marker genes were used to calculate the UMAP, with n.neighbors = 30L and min.dist = 0.3.

### FISH to scRNA-seq data mapping

To map FISH data to scRNA-seq clusters, we first z-score normalized the scRNA-seq data and calculated the average marker gene expression in each scRNA-seq cluster. Marker gene expression measured via EASI-FISH was also z-score normalized. Pearson's correlation was computed between EASI-FISH neurons and scRNA-seq cluster averages. Each EASI-FISH neuron was assigned the scRNA-seq cluster identity with the highest correlation. We then calculated the fraction of neurons from each EASI-FISH cluster that mapped to each scRNA-seq cluster and assigned correspondence based on the highest fraction of neurons from EASI-FISH cluster that mapped to scRNA-seq clusters. MC-7, MC-8, MC-20, and MC-21 were excluded from this analysis as most neurons from these clusters were not in the CEA.

### Spatial parcellation in the CEA and surrounding area

As described before (*Wang et al., 2021*), PCA was performed on the expression patterns of 29 marker genes from each sample to identify the most variable and highly correlated spatial patterns. Marker gene expression from EASI-FISH data were first z-score normalized and mapped in 3D to reconstruct the expression patterns. Images containing expression patterns of 29 marker genes were decomposed into PCs. The eigen images from the top four PCs explained on average 71.3 ± 4.3% of variance in each sample. We selected genes that on average have the largest magnitude of weight on each PC as candidates for structural parcellation. Based on this criterion, *Penk*, *Ppp1r1b*, and *Nefm* were chosen for parcellation in all samples. Neighboring brain regions (BLA, ITCs, PAL, AST) were prominent from the PCA, the boundaries were also confirmed with cell size, soma morphology, and density.

To parcellate the CEA into subregions, neurons were first classified based on their expression of *Penk*, *Ppp1r1b,* and *Nefm*. Otsu's method implemented in Python scikit-image was used to identify the optimal threshold for this classification. Then Gaussian process classifier with the radial basis function (RBF) kernel (length scale: 1.0) was used (implementation in scikit-learn) to generate the probabilistic segmentation (1 µm isotropic resolution). Probability greater than 0.5 was used as threshold for the segmentation mask. This segmentation was performed separately on each CEA sample. Analysis on spatial distribution of molecularly defined cell types were the same as described previously (*Wang et al., 2021*).

### Logistic regression to predict projection class with gene expression

Prediction of projection classes from marker gene expression was assessed using binomial logistic regression. Due to the small number of neurons (1%) with collateral projections (*Figure 5—figure supplement 1C*), these neurons were excluded from this analysis. Neuronal projection types were binarized for each projection class, and z-score normalized gene expression matrix was used as input to train a multiclass logistic regression classifier to predict its projection probability for each target. Logistic regression was implemented using the scikit-learn package linear_model.LogisticRegressionCV() class with liblinear solver and L2 regularization. To account for data imbalance, the class weights were adjusted inversely proportional to class frequencies from the input data. The performance scores were calculated with 20 randomized repeats of stratified fivefold cross-validation. p-value was calculated with permutation test, where the projection classes were shuffled 100 times to generate randomized data to compute the empirical p-value against the null hypothesis that projection classes and marker gene expression are independent.

## Acknowledgements

We thank the following people and groups for their support. J Clements: consultation and help on web portal; T Wang: consultation on experimental procedure; C Stringer: consultation on data analysis; D Alcor, M DeSantis: microscopy support; A Hu, M Copeland, SC Michael, KM McGowan: animal perfusion and tissue sectioning; Janelia Vivarium team: animal care. This work was conducted as part

of the multiFISH Project Team at Janelia Research Campus and research funding was from Howard Hughes Medical Institute (HHMI). SK was supported by a Career Development Award from Dementia Research Switzerland – Synapsis Foundation and the Janelia Research Campus Visitor Program.

## Additional information

### Funding

| Funder | Grant reference number | Author |
| --- | --- | --- |
| Howard Hughes Medical Institute | | Scott M Sternson |
| Dementia Research Switzerland | | Sabine Krabbe |

The funders had no role in study design, data collection and interpretation, or the decision to submit the work for publication.

### Author contributions

Yuhan Wang, Conceptualization, Resources, Data curation, Software, Formal analysis, Validation, Investigation, Visualization, Methodology, Writing – original draft, Writing – review and editing; Sabine Krabbe, Conceptualization, Formal analysis, Investigation, Visualization, Methodology, Writing – review and editing; Mark Eddison, Investigation, Methodology, Writing – review and editing; Fredrick E Henry, Andrew L Lemire, Formal analysis, Investigation; Greg Fleishman, Software, Methodology; Lihua Wang, Investigation; Wyatt Korff, Supervision, Funding acquisition, Project administration; Paul W Tillberg, Conceptualization, Resources, Supervision, Funding acquisition, Methodology, Project administration, Writing – review and editing; Andreas Lüthi, Conceptualization, Supervision, Funding acquisition, Writing – review and editing; Scott M Sternson, Conceptualization, Resources, Supervision, Funding acquisition, Writing – original draft, Project administration, Writing – review and editing

### Author ORCIDs

Yuhan Wang  http://orcid.org/0000-0003-4447-5043
Sabine Krabbe  http://orcid.org/0000-0002-5735-469X
Mark Eddison  http://orcid.org/0000-0003-0280-8137
Andrew L Lemire  http://orcid.org/0000-0002-0624-3789
Wyatt Korff  http://orcid.org/0000-0001-8396-1533
Paul W Tillberg  http://orcid.org/0000-0002-2568-2365
Scott M Sternson  http://orcid.org/0000-0002-0835-444X

### Ethics

All methods for animal care and use were conducted according to National Institutes of Health guidelines for animal research and approved by the Institutional Animal Care and Use Committee (IACUC) at Janelia Research Campus (Protocol number: 18-174).

### Decision letter and Author response

Decision letter https://doi.org/10.7554/eLife.84262.sa1
Author response https://doi.org/10.7554/eLife.84262.sa2

## Additional files

### Supplementary files

• Supplementary file 1. Differentially expressed genes in molecularly defined clusters identified from scRNA-seq data. Gene names are listed in the first column, and molecular clusters in the last column. avg_logFC: log fold-chage of the average expression between cells in the cluster of interest and all other cells; pct.1: the percentage of cells where the feature is detected in the cluster of interest; pct.2: the percentage of cells where the feature is detected in all other cells; p_val_adj: adjusted p-value is based on the Bonferroni correction.

• Supplementary file 2. Marker-genes used for expansion-assisted iterative fluorescence in situ

hybridization (EASI-FISH) experiment in the central nucleus of the amygdala (CEA).

• Supplementary file 3. Summary of cells analyzed in expansion-assisted iterative fluorescence in situ hybridization (EASI-FISH).

• Supplementary file 4. Co-expression relationships of previously studied marker-genes in the CEA subregions based on EASI-FISH. Percent of neurons that co-express selected marker genes (*Prkcd*, *Sst*, *Crh*, *Tac2*, *Nts*, *Pdyn*, *Drd1*, *Drd2*, *Penk*, *Tac1*, *Cartpt*, *Cyp26b1*, *Nefm*) are shown, with darker red color representing a higher percent of co-expression.

• Supplementary file 5. Summary of statistical analyses.

• MDAR checklist

### Data availability

All EASI-FISH data generated in this study have been deposited at figshare (https://doi.org/10.25378/janelia.21171373). We also provide an interactive data portal for data visualization at http://multifish-data.janelia.org/. The single-cell RNA-seq dataset generated in this study has been deposited to GEO (Gene Expression Omnibus, https://www.ncbi.nlm.nih.gov/geo/) with accession number GSE213828. Any additional information required to reanalyze the data reported in this paper is available from the lead contacts upon request.

The following datasets were generated:

| Author(s) | Year | Dataset title | Dataset URL | Database and Identifier |
|---|---|---|---|---|
| Krabbe S, Henry FE, Lemire AL, Wang L, Sternson SM | 2022 | Single cell RNA-sequencing in the mouse central amygdala (CEA) | https://www.ncbi.nlm.nih.gov/geo/query/acc.cgi?acc=GSE213828 | NCBI Gene Expression Omnibus, GSE213828 |
| Wang Y, Krabbe S, Eddison M, Korff W, Tillberg PW, Lüthi A, Sternson SM | 2022 | EASI-FISH reveals spatial and axonal projection patterns of molecularly defined cell types in the central amygdala (CEA) | https://doi.org/10.25378/janelia.21171373 | Figshare, 10.25378/janelia.21171373 |

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
