## [Editor Report]

This study presents a valuable and comprehensive analysis of the molecular identity of neuronal subtypes of the central amygdala, along with their spatial, morphological, and connectivity properties. The evidence supporting the authors' conclusions is compelling and includes the use of rigorous state-of-the-art methodologies for RNA sequencing and spatial profiling as well as a novel approach for integrating molecular identity and axonal projections. This study will interest neuroscientists studying the function of the central amygdala.

---

## [Decision Letter]

**Decision letter after peer review:**

Thank you for submitting your article "Multimodal mapping of cell types and projections in the central nucleus of the amygdala" for consideration by *eLife*. Your article has been reviewed by 3 peer reviewers, and the evaluation has been overseen by a Reviewing Editor and Catherine Dulac as the Senior Editor. The following individual involved in the review of your submission has agreed to reveal their identity: Richard D Palmiter (Reviewer #2).

Essential revisions:

Having consulted with reviewers and editors, our collective impression is that this is a strong study that offers valuable insight into the molecular organization of the central amygdala. However, various points were raised during our discussions that will require your attention before the manuscript can be considered further for publication in *eLife*.

1) The review feels that the current sample size of 1300 cells for scRANseq is quite a low quantity despite having a high sequencing depth. We think that there may be a sampling bias inherent to the low quantity of cells. We request that the authors increase their sample size.

2) The authors should provide a solid justification for excluding some of their clusters, especially in light of the fact that some of them (c1, c4) are rather large clusters for which top marker genes are available in the supplementary data.

3) The authors should fully address the scholarship issues raised by Reviewer #3, as well as minor comments included throughout the reviews that don't require additional experiments.

4) Please note that while Reviewer #3 consider it important that female samples be included in the study, we collectively decided that the additional time and material efforts required to address this point goes beyond the expected value that this information will add to the study. As such, we are not requesting that the authors address this experimentally, but rather that they highlight the male-only samples as a limitation of the study.

*Reviewer #1 (Recommendations for the authors):*

1. Given that sequencing clusters are typically listed in order of the cluster size, c1 and c4 would represent some of the larger clusters in the dataset, and do appear to have top marker genes from the supplementary table. If so, excluding these from spatial validation lacks justification, particularly when the cell number is already quite low. The authors should consider selecting some of these markers from these clusters for spatial validation.

2. The spatial resolution and utility of the EASI-fish technique is showcased elegantly in the present study. However, snRNAseq is also a high throughput technique that should reflect the level of heterogeneity in gene expression at the transcriptomic level. There seems to be a quite a difference in the number of clusters between the two techniques (13 seq-cs and 21 MCs), which the authors do not really discuss. It is unclear whether this is due to the lack of cells in the scRNAseq dataset (1300) compared to the EASI-FISH data set (~33000). This may be addressed by adding more cells to the scRNAseq dataset.

3. The violin plots in Figure 1B and Figure 3B do not appear to be organized by cluster and are hard to interpret in this way. Reordering these based genes that are differentially expressed in each cluster would be a more helpful visualization or even perhaps a gene heatmap, where genes that are more expressed in a certain cluster are placed together.

4. Using EASI-FISH, the authors provide a nice characterization of the morphological properties of different CeA molecular cell types. It might be useful to know how these different morphological properties align with the transcriptomic sequencing clusters and whether there are differentially expressed genes associated with differences in membrane properties.

*Reviewer #2 (Recommendations for the authors):*

1. The authors used a nice approach of injecting the PBN with AAV expressing a fluorescent protein to identify the CEA and facilitate dissection of just that region. Figure S1 (A) shows a low-power picture of the GFP fluorescence, but the boundaries of the CEA are not clear in this low-magnification picture. A higher magnification of the CEA region would help.

2. For the scRNA-Seq experiments it would be useful to include the number of neurons sequenced and the number of reads per cell in the main text or the legend in Figure 1. Also, it would be useful to know the relative abundance of each of the 13 scRNA-Seq UMAP clusters.

3. Some readers might be interested to know that Ppp1r1b encodes DARPP32

4. Although it is described well in methods, on line 148, where authors describe injecting 5 major projection regions of CEA neurons with "separate retrograde tracers." It would be more informative to make it clear that each tracer had a distinct fluorophore.

5. Adding a sentence to the text explaining how the 13 CEA clusters defined by sequencing (Figure 1) became 17 CEA clusters based on molecular markers (Figure 3) would be helpful. Also, MC is never defined: does it mean "molecular clusters" as suggested on line 195?

6. Line 225, "Most CEA neurons have medium size" relative to what?

7. Line 335, It is not clear why the authors chose to emphasize MC2 and MC3 in Figure 5B, C; both express Sst and Pdyn and project to the PBN, but MC2 expresses Nts, Crh, Tac2, and Vipr2 with a small projection, while MC3 with a large projection does not. Adding a concluding sentence would help clarify the choice of illustrating these two clusters. Comparing MC3 in lateral CEA with MC16 in medial CEA, both of which have major projections to PBN would be more interesting.

8. It would be nice to acknowledge the role of the PBN in processing painful stimuli more explicitly as well as the role of CGRP signaling. Some authors refer to the CeC as the "nociceptive amygdala." Some relevant papers are Han..Neugebauer (2005) ; J. Neurosci. 25, 10717; Okutsu… Kato, (2017) Mol Pain 13, 1; Wilson… Carrasquillo, (2019) Cell Rep, 29, 332; Allen…Kolber (2022), Biol. Psych. In press PMID: 32730859.

9. Figure S5 (D) It would be helpful to label the 3 columns "anterior, middle and posterior" if that is what is meant. Comparing animals 1 and 2 as shown in Figure S5 (D) would be more useful than showing panel B, which is not readable.

10. Figure S5 (A) is not readable. It would be sufficient to show only part B and showing ANM#1 (A) next to ANM#2 (A), would make the comparison easier.

11. In Figure S7 (A), Are the relative expression dots based on the average of animals 1 and 2? In panel B, the results for most probes in animals 1 and 2 are similar but Npy1r and Drd1 appear to be underrepresented in animal 1 compared to animal 2. Is there an explanation for that?

*Reviewer #3 (Recommendations for the authors):*

Most of my suggestions are related to scholarship. The only additional experiment that I think is needed is the inclusion of female subjects.

1. Multiple studies have shown sex differences in basic neurobiological findings. However, the experiment in this manuscript was only performed in male subjects. This study would benefit from evaluation of female subjects as it is now standard in the field.

2. The work presented in this manuscript is a purely descriptive genetic and anatomical study that can be applied to the multiple functions of the CeA. While the authors mentioned some of these functions in their introduction, many important CeA-dependent functions that are also cell-type-specific were not mentioned. These include alcohol use (Torruella-Suarez et al. 2020 and Domi et al. 2021), drug craving (Venniro et al. 2018 and 2020), itch (Samineni et al. 2021), general anesthetics (Hua et al. 2020), and pain (Wilson et al. 2019, various papers on CRH cells from the Neugebauer lab). Including the multiple functions of the CeA, many of which have been reported to depend on the same genetically defined cell type, underscores the need for a more detailed categorization of CeA neurons that goes beyond the expression of a single gene.

3. Related to the point above, multiple papers from the Sheets lab have also shown distinct electrophysiological properties of CeA neurons based on subnuclei within the CeA, efferent projections, and morphology. Different CeA subnuclei have also been shown to be functionally distinct in the modulation of multiple behaviors (Kim et al. 2017). Separate work from the Palmiter lab has also shown that the anterior and posterior CeA are also functionally distinct. Including this information in the manuscript will strengthen the justification for the need to identify new strategies to target and manipulate anatomically distinct CeA neurons.

4. The authors mention one study that has previously described single nucleus RNA sequencing in the CeA of rats and present their findings in the context of that previous study. However, at least 2 previously published studies (Samineni et al. 2021 and O'Leary et al. 2022) have also completed transcriptomic analyses of CeA neurons in mice, and these studies were not discussed. Two additional papers from the Tonegawa and Ressler's labs have also completed thorough anatomical descriptions of selected genes within the CeA, including co-expression patterns. Multiple papers have also described the genetic identity of CeA neurons with specific long-range projection targets, including some of the projections described in this manuscript. A transparent discussion of the previously published work and the author's interpretation of the similarities and differences between the present results and previously published studies would be useful. A clear distinction between new findings and results that confirm previously published work is imperative.

5. The authors claim that they evaluated the morphological properties of the molecularly defined CEA cell types but only included size and cell body shape. They concluded that "most neurons in the CEA have a medium size, with a similar cell body shape". This overly simplified conclusion can be misleading, particularly in light of multiple studies demonstrating that CeA neurons are morphologically heterogeneous.

---

## [Author Response]

Essential revisions:Having consulted with reviewers and editors, our collective impression is that this is a strong study that offers valuable insight into the molecular organization of the central amygdala. However, various points were raised during our discussions that will require your attention before the manuscript can be considered further for publication in eLife.1) The review feels that the current sample size of 1300 cells for scRANseq is quite a low quantity despite having a high sequencing depth. We think that there may be a sampling bias inherent to the low quantity of cells. We request that the authors increase their sample size.

We have addressed the reviewer’s concern by combining our original dataset (1,393 cells) with a recently published study (O'Leary et al., 2022) (829 cells) through the canonical-correlation analysis (CCA) implemented in Seurat (Figure 1—figure supplement 3) for a total of 2,222 cells. Dimensionality reduction and clustering analysis on the integrated data showed that all neurons from the O’Leary et al. dataset mapped to neurons from our dataset and no O’Leary et al. specific clusters were identified (Figure 1—figure supplement 3A-B), suggesting that the original dataset covered the molecular diversity in the CEA. We also evaluated the clustering by comparing the integrated clusters (Figure 1—figure supplement 3C) with our original clusters using Jaccard similarity index. We found consistent clustering of most neurons (Figure 1—figure supplement 3D-E), only with the following exceptions: seq-c1 is largely absent from the O’Leary et al. dataset, seq-c4 and seq-c12 were grouped into one cluster (cluster 5), and seq-c13 was split into two clusters, cluster 3 and cluster 6, which are the closest clusters on the cluster tree (Figure 1—figure supplement 3F). An additional 28 neurons in the combined dataset clustered together but showed differential expression of RNA spike-ins (ERCCs) (cluster 1 in the combined data) and represented cells with low quality, as indicated by reduced gene count and total RNA counts (Figure 1—figure supplement 3G). These results indicate that the scRNA-Seq clusters presented in this study does not have a sampling bias.

In addition, we compared our scRNA-Seq clusters with a new preprint (Peters et al., 2022) based on their selected marker gene expressions (Author response image 1 and Author response table 1). Based on examining expression of the marker genes reported by Peters et al. (data not publicly available) in our own scRNA-Seq data, we were able to match most of CEA clusters in the Peters, et al. study to our dataset. We also found additional clusters in our data that were not identified by Peters, et al. For example, *Gal*/*Gpx3* (seq-c3), which we validated by FISH. Notably, Peters et al. sequenced a larger number of cells (3,325 cells) using droplet-based methods, but they did not find additional cell clusters, which further validates our clustering results.

**Author response table 1. sa2table1:** ScRNA-Seq cluster correspondence between the Peters et al. study and this study.

Peters, et al. (9 CeA clusters)	This study	Notes
CeL_Nts.Tac2	seq-c7 (Sst/Pdyn/Crh)	
CeL_Sst	seq-c10 (Sst/Pdyn/Crh--)	
CeM_Vdr	seq-c9 (Vdr)	
CeM_Il1rapl2	seq-c12(Fxyd6/Gnas)	may have included cells from seq-c2, seq-c3 and seq-c4
CeM_Drd2.Rai14	seq-c5 (Drd2/Scn4b)	matched based on Drd2 expression
CeM_Tac1.Sst	seq-c11 (Sst/Tac1/Sema3c)	
CeM_Dlk1	seq-c13 (Cyp26b1/Dlk1)	
CeL_Prkcd	seq-c8 (Prkcd/Cartpt)	
CeC_Cdh9.Calcrl	seq-c6 (Cyp26b1/Crym)	

Taken together, these additional scRNA-Seq studies support the conclusion that our analysis is comprehensive and has minimal sampling bias.

**Author response image 1. sa2fig1:** Comparison of gene expression in identified CEA clusters between Peters et al. dataset (top dot plot) and our dataset (bottom dot plot).

2) The authors should provide a solid justification for excluding some of their clusters, especially in light of the fact that some of them (c1, c4) are rather large clusters for which top marker genes are available in the supplementary data.

We thank the reviewer for raising this point. We have now included additional information in the main text and supplemental figure to discuss the exclusion of seq-c1, seq-c4 and seq-c12 for marker gene selection.

For seq-c1, despite that candidate marker genes were identified from scRNA-Seq data based on their differential expression between seq-c1 and the rest of the population, we found that most of these candidate marker genes showed broad expression patterns in the CEA and surrounding areas, which is also evident in Allen ISH data (Author response image 2). We did not think choosing a gene with such broad regional expression but with expression in only 66% of the neurons of our seq-c1 was useful, and this reduced our confidence in the cluster, so we excluded it. Moreover, in light of the availability to compare our dataset with two new datasets (O’Leary et al. and Peters et al. datasets), this decision to exclude the cluster in marker gene selection appears justified by the fact that seq-c1 was not evident in these two new datasets (Author response table 1 and Figure 1—figure supplement 3B).

For seq-c4 and seq-c12, the candidate marker genes (*Ppia, Tuba1b, Gnas*) lack high cluster selectivity/specificity, based on the fraction of neurons expressing these markers in other clusters (Table S1, all marker genes have pct.2 greater than 0.3 and Author response image 2). Given the high sequencing depth for these neurons, it is unlikely that we have failed to detect highly selective marker genes from these neurons. Instead, these clusters are *Nefm* (*Fxyd6*)*-*expressing CEA GABAergic cells (new Figure 1B) that are mostly distinguished from other clusters by the absence of additional unique marker genes (e.g., lack of *Penk*, *Ppp1r1b* and *Prkcd*, Author response image 2). We included *Nefm* in our panel of marker gene probes. It is interesting to note that, based on FISH data from the recent O’Leary et al. paper, *Fxyd6* expression appears less selective for the CeM than *Nefm* (Author response image 2). Furthermore, clustering analysis after merging our data with O’Leary et al. showed that seq-c4 and seq-c12 were grouped into a single cluster (integrated Cluster 5) that is *Nefm*-expressing and continued to lack highly selective marker genes (Figure 1—figure supplement 3E). This cluster most likely corresponds to CeM_Il1rapl2 in Peters et al. dataset, which also showed the least marker gene selectivity in our scRNA-Seq dataset (Author response image 1). Therefore, we conclude that cells lacking other marker genes (*Penk* and *Ppp1r1b*) but expressing *Nefm* are representative of a corresponding large number of cells in our original scRNA-Seq Clusters 4 &12 (integrated Cluster 5).

Thus, based on evidence above, we did not include additional differentially expressed (DE)-genes identified from seq-c1, seq-c4, and seq-c12 for further analysis.

**Author response image 2. sa2fig2:** Candidate marker genes and their expression from seq-c1, c4 and c12. (A) Sagittal sections from Allen ISH database showing expression patterns of selected seq-c1 marker genes at the central amygdala and surround area, as schematized from reference atlas-defined regions. (B) Violin plot showing expression of selected seq-c4 and seq-c12 marker genes in scRNA-Seq clusters. Gene expression is first normalized by the total counts in that cell, multiplied by 10,000 and then natural log transformed. (C) Left: Sagittal sections from Allen ISH database showing expression patterns of selected seq-c4 marker genes at the central amygdala and surrounding area, as schematized from reference atlas-defined regions. Right: Coronal sections from Allen ISH database showing expression patterns of selected seq-c12 marker genes at the central amygdala and surrounding area, as schematized from reference atlas-defined regions. Scale bars in A and C: 840 µm. (D) Fxyd6 expression in the central amygdala, as detected by FISH. Duplicated from O’Leary et al. study. (E) Volcano plots showing differentially expressed genes in seq-c4, seq-c12 and combined (seq-c4 & seq-c12) clusters. Genes with log2 fold change greater than 0.75 and adjusted p-value less than 1e-5 are labeled.

3) The authors should fully address the scholarship issues raised by Reviewer #3, as well as minor comments included throughout the reviews that don't require additional experiments.

We thank the reviewer for raising this very important point. These issues and comments have all been addressed.

4) Please note that while Reviewer #3 consider it important that female samples be included in the study, we collectively decided that the additional time and material efforts required to address this point goes beyond the expected value that this information will add to the study. As such, we are not requesting that the authors address this experimentally, but rather that they highlight the male-only samples as a limitation of the study.

We thank the reviewer for this comment, and we have now included a section in the discussion to highlight the limitation of this study with the male-only samples (Please also see response to Reviewer #3).

Reviewer #1 (Recommendations for the authors):1. Given that sequencing clusters are typically listed in order of the cluster size, c1 and c4 would represent some of the larger clusters in the dataset, and do appear to have top marker genes from the supplementary table. If so, excluding these from spatial validation lacks justification, particularly when the cell number is already quite low. The authors should consider selecting some of these markers from these clusters for spatial validation.

Please see above. Most marker genes identified in these clusters lack specificity.

2. The spatial resolution and utility of the EASI-fish technique is showcased elegantly in the present study. However, snRNAseq is also a high throughput technique that should reflect the level of heterogeneity in gene expression at the transcriptomic level. There seems to be a quite a difference in the number of clusters between the two techniques (13 seq-cs and 21 MCs), which the authors do not really discuss. It is unclear whether this is due to the lack of cells in the scRNAseq dataset (1300) compared to the EASI-FISH data set (~33000). This may be addressed by adding more cells to the scRNAseq dataset.

We thank the reviewer for raising the point and we have included in the main text (Page 11, Line 243) to discuss the potential discrepancy between scRNA-Seq (13 CEA clusters) and EASI-FISH clusters (17 CEA clusters + 4 clusters from surrounding brain region). We agree with the reviewer that this could be related to the difference in number of cells between the two datasets, as all additional EASI-FISH clusters are further split from existing scRNA-Seq clusters. For example, seq-c3 (Gpx3/Gal) cluster corresponds to MC-9 (Gpx3/Gal) and MC-19 (Gpx3/Gal-). And Gpx3/Gal- cells are observed in seq-c3 cluster. However, adding more cells to the scRNAseq dataset (see above) did not dramatically change the clustering of scRNAseq data (Figure 1—figure supplement 3), suggesting that current scRNAseq dataset is sufficient to cover the molecular diversity in the CEA.

3. The violin plots in Figure 1B and Figure 3B do not appear to be organized by cluster and are hard to interpret in this way. Reordering these based genes that are differentially expressed in each cluster would be a more helpful visualization or even perhaps a gene heatmap, where genes that are more expressed in a certain cluster are placed together.

We thank the reviewer for the suggestion and have now replaced Figure 1B and Figure 3B with heatmaps. Genes are now sorted based on their expression in different clusters.

4. Using EASI-FISH, the authors provide a nice characterization of the morphological properties of different CeA molecular cell types. It might be useful to know how these different morphological properties align with the transcriptomic sequencing clusters and whether there are differentially expressed genes associated with differences in membrane properties.

We have now included the top differentially expressed genes encoding membrane proteins in clusters that showed different morphological properties (seq-c2 that correspond to MC-1 and seq-c3 that correspond to MC-9) in Figure 3—figure supplement 1F-G.

Reviewer #2 (Recommendations for the authors):1. The authors used a nice approach of injecting the PBN with AAV expressing a fluorescent protein to identify the CEA and facilitate dissection of just that region. Figure S1 (A) shows a low-power picture of the GFP fluorescence, but the boundaries of the CEA are not clear in this low-magnification picture. A higher magnification of the CEA region would help.

Thank you for the suggestion. We have now included zoomed-in views before and after the CEA dissection in Figure 1—figure supplement 1.

2. For the scRNA-Seq experiments it would be useful to include the number of neurons sequenced and the number of reads per cell in the main text or the legend in Figure 1. Also, it would be useful to know the relative abundance of each of the 13 scRNA-Seq UMAP clusters.

We thank the reviewer for the suggestion, and we have now included the requested information in the legend in new Figure 1. The relative abundance of each cluster is now shown in new Figure 1D.

3. Some readers might be interested to know that Ppp1r1b encodes DARPP32

This has been added to the main text (Page 6, Line 118).

4. Although it is described well in methods, on line 148, where authors describe injecting 5 major projection regions of CEA neurons with "separate retrograde tracers." It would be more informative to make it clear that each tracer had a distinct fluorophore.

This has been clarified in the main text (Page 8, Line 174).

5. Adding a sentence to the text explaining how the 13 CEA clusters defined by sequencing (Figure 1) became 17 CEA clusters based on molecular markers (Figure 3) would be helpful. Also, MC is never defined: does it mean "molecular clusters" as suggested on line 195?

Both points have been clarified in the main text Point 1: Page 10, Line 234 and Page 11, Line 243; Point 2: Page 10, Line 237.

6. Line 225, "Most CEA neurons have medium size" relative to what?

We have now modified the language in the main text to avoid confusion.

7. Line 335, It is not clear why the authors chose to emphasize MC2 and MC3 in Figure 5B, C; both express Sst and Pdyn and project to the PBN, but MC2 expresses Nts, Crh, Tac2, and Vipr2 with a small projection, while MC3 with a large projection does not. Adding a concluding sentence would help clarify the choice of illustrating these two clusters. Comparing MC3 in lateral CEA with MC16 in medial CEA, both of which have major projections to PBN would be more interesting.

We have now included the requested figures (new Figure 5C) and clarified the comparison between MC-2 and MC-3 (new Figure 5D) in the main text (Page 16, Line 383).

8. It would be nice to acknowledge the role of the PBN in processing painful stimuli more explicitly as well as the role of CGRP signaling. Some authors refer to the CeC as the "nociceptive amygdala." Some relevant papers are Han..Neugebauer (2005) ; J. Neurosci. 25, 10717; Okutsu… Kato, (2017) Mol Pain 13, 1; Wilson… Carrasquillo, (2019) Cell Rep, 29, 332; Allen…Kolber (2022), Biol. Psych. In press PMID: 32730859.

We now discuss the role of PBN-CEA circuit in pain and aversion and also included the mentioned references (Page 4 Line 64).

9. Figure S5 (D) It would be helpful to label the 3 columns "anterior, middle and posterior" if that is what is meant. Comparing animals 1 and 2 as shown in Figure S5 (D) would be more useful than showing panel B, which is not readable.

We have now added the labels and revised the figures to compare animal 1 and animal 2 in Figure 4—figure supplement 2.

10. Figure S5 (A) is not readable. It would be sufficient to show only part B and showing ANM#1 (A) next to ANM#2 (A), would make the comparison easier.

We have now revised the figures to clarify and also show ANM#1 next to ANM#2 for comparison (Figure 4—figure supplement 1).

11. In Figure S7 (A), Are the relative expression dots based on the average of animals 1 and 2? In panel B, the results for most probes in animals 1 and 2 are similar but Npy1r and Drd1 appear to be underrepresented in animal 1 compared to animal 2. Is there an explanation for that?

Figure S7A is based on the averaged expression of animals 1 and 2. *Npy1r* and *Drd1* are low expressing genes that were detected in the same round of FISH and slight difference in background spots in animal 2 for these two genes (that were probed in the same round of FISH) led to the color differences in these heatmaps. To improve the clarity of the display, we now z-score normalized the spot counts and replotted Figure S7 (now Figure 4—figure supplement 4). We have also confirmed these results in animal 3.

Reviewer #3 (Recommendations for the authors):Most of my suggestions are related to scholarship. The only additional experiment that I think is needed is the inclusion of female subjects.1. Multiple studies have shown sex differences in basic neurobiological findings. However, the experiment in this manuscript was only performed in male subjects. This study would benefit from evaluation of female subjects as it is now standard in the field.

We thank the reviewer for bringing up this important point and we now discuss this limitation in the main text (Page 21, Line 493). Additionally, we also compared our data with another published scRNA-Seq dataset (Peters et al., 2022) that included cells from both male and female animals (Author response table 1) and found correspondence of all clusters in our dataset. Of note, Peters et al. study also did not report any sex differences in their scRNA-Seq clusters.

2. The work presented in this manuscript is a purely descriptive genetic and anatomical study that can be applied to the multiple functions of the CeA. While the authors mentioned some of these functions in their introduction, many important CeA-dependent functions that are also cell-type-specific were not mentioned. These include alcohol use (Torruella-Suarez et al. 2020 and Domi et al. 2021), drug craving (Venniro et al. 2018 and 2020), itch (Samineni et al. 2021), general anesthetics (Hua et al. 2020), and pain (Wilson et al. 2019, various papers on CRH cells from the Neugebauer lab). Including the multiple functions of the CeA, many of which have been reported to depend on the same genetically defined cell type, underscores the need for a more detailed categorization of CeA neurons that goes beyond the expression of a single gene.

We thank the reviewer for the comments and have now included these important functional aspects and corresponding references in the introduction section (Page 3-5).

3. Related to the point above, multiple papers from the Sheets lab have also shown distinct electrophysiological properties of CeA neurons based on subnuclei within the CeA, efferent projections, and morphology. Different CeA subnuclei have also been shown to be functionally distinct in the modulation of multiple behaviors (Kim et al. 2017). Separate work from the Palmiter lab has also shown that the anterior and posterior CeA are also functionally distinct. Including this information in the manuscript will strengthen the justification for the need to identify new strategies to target and manipulate anatomically distinct CeA neurons.

We thank the reviewer for the comments and have now included these references in the main text (Page 4, last paragraph).

4. The authors mention one study that has previously described single nucleus RNA sequencing in the CeA of rats and present their findings in the context of that previous study. However, at least 2 previously published studies (Samineni et al. 2021 and O'Leary et al. 2022) have also completed transcriptomic analyses of CeA neurons in mice, and these studies were not discussed. Two additional papers from the Tonegawa and Ressler's labs have also completed thorough anatomical descriptions of selected genes within the CeA, including co-expression patterns. Multiple papers have also described the genetic identity of CeA neurons with specific long-range projection targets, including some of the projections described in this manuscript. A transparent discussion of the previously published work and the author's interpretation of the similarities and differences between the present results and previously published studies would be useful. A clear distinction between new findings and results that confirm previously published work is imperative.

We have now included extensive additional information and detailed analysis in the results and figures to address the reviewer’s suggestion for O’Leary et al. 2022. Samineni et al. 2021 was cited by us, but this study uses bulk RNA-Seq and was less relevant to the analyses that we performed here. We cited the influential two papers from Kim/Tonegawa (14 times) and McCullough/Ressler (4 times) extensively in the context of our analysis of cell types, which serves to point the reader to past quantitative work examining the co-expression of subsets of the marker genes evaluated by us. For example, we note in the Introduction: “Additional work has demonstrated that some of these markers (Sst, Crh, Nts) have a considerable degree of overlap in the same neurons, whereas others are distinct (e.g., Prkcd) (McCullough, Morrison, et al., 2018).” However, we defined cell populations in the context of co-expression of many more genes, thus it is difficult to perform a detailed comparison. We state this in the discussion: “Although CeM neurons have been reported to be associated with neuropeptide marker genes (Kim et al., 2017; McCullough, Morrison, et al., 2018), we found that these neuropeptide-expressing cell types made up only a small proportion of the cells in the CeM.” We also mention that: “Other previously reported CeM markers were expressed in more than one molecularly defined cell type. For example, Pnoc (Hardaway et al., 2019) was broadly expressed across most major CeM cell types (Figure 1—figure supplement 2). While the D2-receptor (Drd2) was primarily expressed in two CeM clusters (MC-5 and MC-14, both corresponding to seq-c5 but differing in Penk expression), the D1-receptor (Drd1) was primarily expressed in the AST cluster (MC-7) and the intercalated cells (MC-21). Drd1 was also detected (>10%) in multiple CeM clusters (MC-1, MC-11, MC-12, MC-16, and MC-18: seq-c2, seq-c4, seq-c11) (Figure 1—figure supplement 2 and Figure 4—figure supplement 4), largely consistent with a previous report (Kim et al., 2017).” In the results we note: “Although calcitonin receptor-like (Calcrl) has been proposed as a distinguishing marker gene between Prkcd neuron types (Kim et al., 2017), it was expressed in a subset of cells in both seq-c6 (9.1% of neurons) and seq-c8 (36% of neurons).” Related to this, in the Discussion we note: “We identified two populations with Prkcd expression, one in rostral CEA (MC-10 corresponds to seq-c6) and one in caudal CEA (MC-6 corresponds to seq-c8) with a higher fraction of neurons expressing Prkcd, which likely correspond to the rCEA Calcrl+ and cCEA Calcrl+ neurons identified in (Bowen et al., 2022).”Also for CeL and CeC we noted: “Consistent with previous reports, we detected significant levels of co-localization in CeC/CeL of previously used marker genes such as Sst, Crh, Nts and Tac2, which were distinct from Prkcd-expressing neurons (Kim et al., 2017; McCullough, Morrison, et al., 2018; Ye and Veinante, 2019).” And “A closely related Sst-expressing cell type, MC-2, expressed Vipr2, was partially offset to the medial portion of the CeL and co-expressed varying proportions of Sst, Nts, Crh (Kim et al., 2017; McCullough, Morrison, et al., 2018).” And “The anterior CeC also contained a Drd2 cell type co-expressing Scn4b (MC-5) that extended ventrally from the AST and projected to lateral SN, which likely corresponded to previously reported Drd2 neurons in and around CEA that enhance conditioned freezing (McCullough, Daskalakis, et al., 2018).” We also wrote: “Recently, single nucleus RNA sequencing from the CEA in rat identified 13 neuronal types (11/13 neuronal types were from CEA) (Dilly et al., 2022). We found good correspondence with major CEA cell types identified by Dilly, et al., such as the Prkcd and the Crh neurons in the CeC and CeL, and the Drd1 and Drd2 neurons in the CeM. However, the separation of rat CEA Sst subtypes in Dilly, et al. was not clear. In addition to these major cell types, our study revealed additional cell type diversity in the mouse CEA, especially in the CeM, which had been largely underdefined. In addition, our molecularly defined mouse CEA cluster also matched well with two recent studies in mouse CEA (O'Leary et al., 2022; Peters et al., 2022) (Figure 1—figure supplement 3).”

Moreover, the incidence of detected genes and co-expression values (e.g. % positive) will be sensitive to differences in analysis, which was based on transcript count detections for us and integrated fluorescence intensity for Tonegawa and Ressler studies. Slight differences in how experimenters defined their baseline values can influence the number of cells reported as positive for gene. In our case, we thresholded the gene expression against background spots and any cell with a spot count of greater than 10 was identified as gene positive.

We have cited a number of cases of analysis of projections from cells expressing specific marker genes: “We did not evaluate intra-CeA connectivity, but Prkcd neurons of the CeC and CeL have been reported to also form connections with CeM neurons (Ye and Veinante, 2019), some of which go on to project to the PAG (Haubensak et al., 2010). In addition, Prkcd-neurons are also engaged in a recursive inhibitory local circuit with CeL Sst neurons (Fadok et al., 2017; Hunt et al., 2017).” We also note in the discussion: “The functional significance of two Sst neuron populations in CeL is not known, but functional activation of CeLCRH neurons, possibly projecting to vlPAG (likely MC-2), has been shown to promote flight responses (Andero et al., 2014), whereas activation of CeLSST projecting to vlPAG (likely MC-2 and MC-3) promotes freezing (Fadok et al., 2017; Kim et al., 2017; Yu et al., 2016). However, these earlier studies may include contributions from neurons expressing these genes in CeM, thus state-dependent effects (Fadok et al., 2018; Moscarello and Penzo, 2022) that primarily affect separate CEA subpopulations cannot be excluded.” We also mention other CEA axon projection studies in the context of our findings: “However, it remains to be investigated whether these distinct molecular clusters with common projection targets serve specific functions, such as associative learning to salient stimuli, feeding or predatory behavior which have been previously linked to CEA projections to the lateral SN, PBN or PCRt, respectively (Douglass et al., 2017; Han et al., 2017; Steinberg et al., 2020).”

5. The authors claim that they evaluated the morphological properties of the molecularly defined CEA cell types but only included size and cell body shape. They concluded that "most neurons in the CEA have a medium size, with a similar cell body shape". This overly simplified conclusion can be misleading, particularly in light of multiple studies demonstrating that CeA neurons are morphologically heterogeneous.

We thank the reviewer for raising this important point and have now modified the language to limit our description to soma and discussed the limitation of this characterization in Page 12 Line 276.